# Structural basis of HLX10 PD-1 receptor recognition, a promising anti-PD-1 antibody clinical candidate for cancer immunotherapy

Hassan Issafras[1]*, Shilong Fan[2]*, Chi-Ling Tseng[3], Yunchih Cheng[3], Peihua Lin[1], Lisa Xiao[4], Yun-Ju Huang[3], Chih-Hsiang Tu[3], Ya-Chin Hsiao[3], Min Li[2], Yen-Hsiao Chen[3], Chien-Hsin Ho[3], Ou Li[1], Yanling Wang[1], Sandra Chen[5], Zhenyu Ji[4], Eric Zhang[4], Yi-Ting Mao[1], Eugene Liu[6], Shumin Yang[4], Weidong Jiang[1,4]

1 Hengenix Inc., Fremont, CA, United States of America, 2 National Protein Science Facility, Tsinghua University, Beijing, China, 3 HanchorBio Inc., Ltd, Taipei, Taiwan, 4 Shanghai Henlius Biotech, Inc., Shanghai, P. R. China, 5 Anwita Biosciences, San Carlos, CA, United States of America, 6 Taipei Medical University, Taipei, Taiwan

* hissafras@gmail.com (HI); fanshilong@mail.tsinghua.edu.cn (SF)

## Abstract

Cancer immunotherapies, such as checkpoint blockade of programmed cell death protein-1 (PD-1), represents a breakthrough in cancer treatment, resulting in unprecedented results in terms of overall and progression-free survival. Discovery and development of novel anti PD-1 inhibitors remains a field of intense investigation, where novel monoclonal antibodies (mAbs) and novel antibody formats (e.g., novel isotype, bispecific mAb and low-molecular-weight compounds) are major source of future therapeutic candidates. HLX10, a fully humanized IgG$_4$ monoclonal antibody against PD-1 receptor, increased functional activities of human T-cells and showed *in vitro*, and anti-tumor activity in several tumor models. The combined inhibition of PD-1/PDL-1 and angiogenesis pathways using anti-VEGF antibody may enhance a sustained suppression of cancer-related angiogenesis and tumor elimination. To elucidate HLX10's mode of action, we solved the structure of HLX10 in complex with PD-1 receptor. Detailed epitope analysis showed that HLX10 has a unique mode of recognition compared to the clinically approved PD1 antibodies Pembrolizumab and Nivolumab. Notably, HLX10's epitope was closer to Pembrolizumab's epitope than Nivolumab's epitope. However, HLX10 and Pembrolizumab showed an opposite heavy chain (HC) and light chain (LC) usage, which recognizes several overlapping amino acid residues on PD-1. We compared HLX10 to Nivolumab and Pembrolizumab and it showed similar or better bioactivity *in vitro* and *in vivo*, providing a rationale for clinical evaluation in cancer immunotherapy.

## Introduction

Programmed cell death 1 (PD-1), also known as CD274 is a co-inhibitory receptor expressed by all T-cells during activation and other immune cells (NK, B cell). Upon binding to its

**Data Availability Statement:** This data is currently published in PDB data base: https://www.rcsb.org/structure/7E9B and there are no legal or ethical restrictions on sharing our data publicly.

**Funding:** The authors received funding in the form of salary for this work from Shanghai Henlius Biotech, Inc. Hassan Issafras, Chi-Ling Tseng, Yunchih Cheng, Peihua Lin, Lisa Xiao, Yun-Ju Huang, Chih-Hsiang Tu, Ya-Chin Hsiao, Yen-Hsiao Chen, Chien-Hsin Ho, Ou Li, Yanling Wang, Sandra Chen, Zhenyu Ji, Eric Zhang, Yi-Ting Mao, Eugen Liu, Shumin Yang and Weidong Jiang were employees of Shanghai Henlius Biotech, Inc., P. R. China and its US subsidiary Hengenix Inc. Fremont, CA, US. Shilong Fan and Min Li received funding from Shanghai Henlius Biotech, Inc. but were not salaried employees. The funders had no role in study design, data collection and analysis, decision to publish, or preparation of the manuscript. No additional external funding was received for this study. The role of the authors as articulated in the 'author contributions' section.

**Competing interests:** The study is about HLX10; a clinical lead currently in development in China and US. Henlius filled patent in US (US2021277122A1). There are no additional patents, products in development or marketed products associated with this research to declare. This work does not alter PLOS's adherence policies on sharing data and materials.

ligands PD-L1 (B7-H1) and PD-L2 (B7-DC), PD-1 regulates T-cell effector functions during various physiological responses, including acute and chronic infection, and the maintenance of immune tolerance [1,2]. Increased expression of PD-L1 in tumor tissues was initially considered a major mechanism of cancer-mediated T-cell immunosuppression and exhaustion [2,3]. Subsequently, it became apparent that PD-L1 expressed in antigen-presenting myeloid cells in the tumor microenvironment is as important for mediating T-cells immunosuppression [4]. PD1 is made of an extracellular immunoglobulin-like binding domain, a transmembrane region and a cytoplasmic domain containing an immunoreceptor tyrosine-based inhibitory motif (ITIM) and an immunoreceptor tyrosine-based switch motif (ITSM) [5,6]. Mechanistically upon PD-L-1 binding to PD-1, it interferes with antigen presentation and T-cell receptor (TCR) signal transduction by recruiting the tyrosine phosphatase SHP2, thereby dephosphorylating proximal signaling elements such as PI3K/AKT and Ras-MEKERK pathways [7]. This dephosphorylation inhibits T-lymphocyte proliferation, release of cytokines, and cytotoxicity, resulting in exhaustion of tumor-specific T-cells and cancer escape. The inhibition of PD-1/PD-L1 pathway using monoclonal antibodies (mAbs) results in the reversal of the exhausted T-cell phenotype and thereby enabling tumor-reactive T-cells to recognize tumor antigens, providing a rationale for cancer immunotherapy [8]. Cancer immunotherapy using mAbs against PD-1 (and its ligand PD-L1) has demonstrated unprecedented therapeutic benefits and helped to provide long-term durable responses in a subset of patients with multiple types of advanced cancers [9,10]. Nivolumab (Opdivo®) and Pembrolizumab (Keytruda®) are the first two anti–PD-1 mAbs that have received US Food and Drug Administration (FDA) approval in several cancers with some overlapping indications e.g., melanoma and non-small cell lung cancer. These two mAbs are both IgG4 subtype antibodies and bind with different affinities to slightly different epitopes in PD-1, as suggested by structural comparative studies [11,12]. The crystal structures of PD-1/PD-L1 [13], PD-1/Pembrolizumab complex [14,15] and PD-1/Nivolumab complex [16] were all reported. Such structural data provided important information about the molecular interaction and thus represent good references for the development of novel and more effective mAbs in the future. More recently, additional anti-PD-1 mAbs were either approved: e.g., Cemiplimab for cutaneous squamous cell carcinoma; Sintilimab approved by the *National Medical Products Administration* (NMPA) for the treatment of relapsed or refractory classic Hodgkin's lymphoma, or currently ongoing clinical evaluation e.g., Tislelizumab and Dostarlimab.

Here, we describe HLX10, a novel fully humanized anti PD-1 IgG$_4$ mAb that demonstrated a pronounced efficacy *in vivo*. We characterize its *in vitro* activity and show that HLX10 activate T-cell proliferation and cytokine secretion in T-cells. Furthermore, HLX10 inhibits tumor growth in several syngeneic and xenograft models and synergizes with Avastin biosimilar to promote robust tumor activity. To gain insight into how HLX10 achieves PD-1 recognition, we determined the co-crystal structure of the antigen-binding fragment (Fab) of HLX10 in complex with PD-1 at a 1.78-Å resolution and compared this structure to the previously determined structures of Pembrolizumab and Nivolumab.

## Materials and methods

### Reagents

Recombinant purified human PD-1 protein, residues Leu25-Thr168, with a C-terminal 6-His tag, recombinant cynomolgus monkey PD-1 His tag (R&D systems #8509-PD-050) were purchased from R&D Systems (catalog# 8986-PD). Recombinant cynomolgus monkey PD-1-ECD-Fc (catalog# 90311-C02H), mouse PD-1-ECD-Fc (catalog# 50124-M02H) and rat PD-1 ECD-Fc (catalog# 80448-R02H) were purchased from Sino Biological Inc. Recombinant

human PD-1-ECD-Fc was expressed in CHO-S cells by pairing human PD-1 (Leu25-Thr168) to human IgG1 Fc and purified with protein A agarose (MabSelect SuRe; GE Healthcare Life Sciences). Recombinant human PDL-1 alkaline phosphatase (PDL-1 AP) was expressed in CHO-S cells by pairing human PDL-1 (19–239) to alkaline phosphatase (20–511) followed by 6xHis Tag and purified with Nickel agarose resin (Qiagen).

HLX10 was expressed in Chinese hamster ovary cell line (CHO-S1) cells as IgG4 isotype with a stabilizing 226 mutation [17]. The nucleotide sequences of the VH and VL encoding gene fragments (GenScript, Piscataway, NJ, USA) were cloned into expression vector (AS-puro from EMD Millipore) and used to generate a stable single clone. Nivolumab (Nivo) analog was generated in house based on the sequences from the World Health Organization-INN literature. The light chain and heavy chain were synthesized by GenScript (New Jersey, United States) and subcloned into the expression vector (AS-puro from EMD Millipore). Antibody protein was produced by stable transfection of CHO-S cells, purified by protein A agarose (MabSelect SuRe; GE Healthcare Life Sciences) and CEX resin (Poros XS50; Thermo Scientific). The final product was buffer exchanged into 20 mM Tris, 100 mM sodium chloride, 1% Mannitol, 0.1 mM pentetic acid, and 0.01% Tween 80, pH 7.0.

## Tumor cell cultures

All cells mentioned in the following studies were incubated at 37°C with an atmosphere of 5% $CO_2$. HT-29 (human colorectal carcinoma) and NCI-H292 (human non-small-cell lung cancer) were purchased from ATCC. HT-29 was cultured in McCoy's 5A media (Corning cellgro) supplemented with 10% FBS (Biological Industries). NCI-H292 was cultured in RPMI-1640 media (Corning cellgro) supplemented with 10% FBS. Cells were passaged 2–3 times per week and incubated at 37°C with an atmosphere of 5% CO2.

## Hybridoma generation, VH/VL cloning and humanization

We immunized mice with the recombinant human PD-1 protein (purified recombinant His-tagged PD-1 ECD antigen) and identified twelve hybridoma binders. Clone 15M1 antibody bound to human PD-1 protein by standard enzyme-linked immunosorbent assay (ELISA) and Flow cytometry and blocked PDL-1 ligand binding. To determine the sequence of clone 15M1, variable regions of heavy (HC) and light chain (LC) were separately amplified with a specific set of primer mixture for murine IgG as previously described [18], and then assembled with nucleotide sequences of protein III of phage and signal peptide to form a long fragment. The resulting fragment was then inserted into phagemid vector and transformed to SS320 *E. coli* to construct a phage display library expressing monovalent Fabs. Colonies of the SS320 cells were picked and cultured in Y2T medium with IPTG to induce Fab secretion. Supernatants with Fab fragments from 96 well plate were screened by ELISA assays using recombinant human PD-1 coated plates. Positive clones by ELISA were sequenced and clones with the same sequence were picked for further characterization. Clone named 1G4 were obtained and confirmed for it binding ability to recombinant human PD-1 protein and blocking ability. To further confirm 1G4 activity, chimeric 1G4 IgG with human Fc fragment (cIG4) was generated and re-tested for binding and blocking of PD-1 PDL-1 interaction.

The humanized anti-PD-1 antibody 1G4 (h1G4) was generated using human germline light chain variable region IGKV1-39*01 and human germline heavy chain variable region IGHV3-11*04. Briefly, humanization was done by grafting the CDR fragment from the light chain and heavy chain of chimeric cIG4 to a similar light chain and heavy chain frameworks of human immunoglobulin using homology search from the publicly disclosed IgGblast database.

Resulting hIG4 was expressed in CHO-S cell, purified by standard protein A resin, and tested for binding to PD-1 and PD-1/PDL-1 blocking ability.

## Recombinant PD-1/PD-L-1 blocking assay

A biochemical PD-1/PDL-1 assay was developed using recombinant human PD-l-His and PD-Ll-AP proteins. Serial dilutions of chimeric clG4 and the Nivolumab analogue antibody were incubated with PD-Ll-AP at RT for 2 hours. Each antibody-antigen mixture was added to PD-l-His-coated wells of a microtiter dish. Following 30 min incubation and wash, p-Nitro-phenyl Phosphate (pNPP) was added to the wells and incubated for 30 minutes. AP activity was measured by monitoring the increase in absorbance at 405 nm. Antibody concentration was plotted as a function of AP activity. The half maximal effective concentration (EC50) was calculated by GraphPad Prism 6 software.

## *In vitro* binding affinity and species cross-reactivity

The recombinant cynomolgus monkey, mouse and rat PD-1 ECD-Fc were purchased from Sino Biological Inc. PD-1-Fc proteins (9 ng per well) were immobilized onto 96-well assay plate (Corning) by incubating overnight at 4˚C. The plate was washed with PBS containing 0.05% Tween 20 and blocked with 5% skim milk in PBS for 1 hour at room temperature. After blocking, serial diluted h1G4 (named HLX10) was added and incubated with the immobilized proteins for one hour at room temperature. The plate was then washed three times with PBST and incubated for one hour at room temperature with peroxidase-labeled goat anti-human IgG F(ab)'$_2$ (Jackson ImmunoResearch Laboratories) diluted 1/10,000 in PBS. The color was developed using TMB substrate (eBioscience) and the absorbance was read at the wavelength of 450 nm by Varioskan Lux (Thermo Fisher Scientific). The half maximal effective concentration (EC50) was calculated by GraphPad Prism 6 software.

Binding by flow cytometry was conducted on CHO-S, PD-1 expressing CHO-s cells, and PHA-activated peripheral blood T-cells using standard flow procedure (S1 File). For T-cell activation, freshly isolated T-cells were activated with PHA (Sigma) at 5 µg/ml for 96 hours. After washing, cells were blocked with FACS buffer (PBS with 2% FBS), containing Fc block (eBioscience), and stained with either HLX10 or Nivo (5 µg//ml), After washing, cells were stained for 30 min at 4˚C with FITC-conjugated rabbit anti-human Fc IgG (Pierce) + CD3-PE antibody (eBioscience) for T-cell gating. After washing twice with FACS buffer, Fluorescence was analyzed on a BD Cytomics FC500 Flow Cytometer (Beckman Coulter).

## Binding kinetics comparison of mAbs by biolayer interferometry

The comparison of HLX10 affinity to human PD-1 to other anti-PD-1 reference mAbs was conducted using BLI method on Octet Red96 instrument (ForteBio). Nivolumab (Opdivo®), Pembrolizumab (Keytruda®), Sintilimab (Tyvyt®) and Toripalimab (JS001) antibody drugs were commercially purchased in their original packages. Briefly, for human PD-1 binding assays, protein A biosensor tips (ForteBio #18–5010) were used to capture anti PD-1 mAbs by dipping the tips in HBS-EP+ (*10 mM HEPES, 150 mM NaCl, 3 mM EDTA and 0.05% P20, pH 7.4)* buffer, containing 5 µg/ml of antibodies. The biosensor tips were then dipped for 120 sec into wells containing a serial dilution of rhPD-1 (R&D systems #8986-PD), followed by a 300 sec dissociation step in HBS-EP+ buffer. All steps were performed at 25˚C and 1000 rpm. Data were globally fit to the 1:1 binding model. The apparent dissociation ($k_d$), association ($k_a$) rate constants and the apparent dissociation equilibrium constant ($K_D$) were calculated with the Octet Data Analysis v11.0 software.

## Cellular PD-L1 and PD-L2 blocking assay

For analyzing the ligand binding blocking of HLX10 on the cell surface, PD-1-transfected CHO-S cells were incubated with serial diluted HLX10 and 5 μg/mL of biotin-labeled PD-L1 or 0.5 μg/mL of biotin-labeled PD-L2 at 4˚C for 30 minutes. Cells were washed with PBS containing 2% FBS twice. Subsequently, PD-1-expressing CHO-S cells were stained with streptavidin-PE (eBioscience). Samples were analyzed by Cytomics FC500 Flow Cytometer (Beckman Coulter).

## PD-1/PD-L1 NFAT reporter bioassay

PD-1/PD-L1 blockade bioassay (Promega) was used following manufacturer's recommendations. PDL1 aAPC/CHO-K1 cells were plated at the density of $4 \times 10^4$ cells/well and cultured overnight in 100 μl of growth medium (Ham's F12 with 10% FBS) in flat-bottom tissue culture plate at 37˚C and 5% $CO_2$. After 24 hours, the growth media was replaced with fresh media. An eight-point dose titration of anti-PD1 antibodies from 0.006 to 100 nM was added to the cells and allowed to bind for 15 min. Subsequently, 40 μl of Jurkat effector cells were mixed and added to each well. The cell mixture was then cultured for 5 hours at 37˚C and 5% $CO_2$. To detect NFAT induced luciferase activity, 80 μl of reconstituted Bio-Glo™ (Promega) was added to each well, and luminescence was recorded using an Envision instrument (Perkin Elmer) after 5 minutes.

## *In vitro* mixed leukocyte reaction assay

Human blood was obtained using Taipei Medical University—Joint Institutional Review Board approved consent forms and protocols. Human peripheral blood mononuclear cells (hPBMCs) were isolated from healthy donors' whole blood by Histopaque®-1077 (Sigma-Aldrich) density centrifugation in SepMate™-50 (STEMCELL Technologies) tubes. Isolated PBMCs were washed and suspended in DPBS. Human T or $CD4^+$ T-cells were isolated from peripheral blood mononuclear cells (PBMCs) using MagniSort™ Human T or $CD4^+$ T-cell Enrichment Kit (eBioscience). Isolated T-cells were cultured in RPMI-1640 medium (Corning cellgro) supplemented with 10% FBS (Biological Industries) and 100 IU/mL penicillin-100 μg/mL streptomycin (1×P/S) (Corning cellgro). To generate dendritic cells (DCs), monocytes isolated from healthy donor PBMCs were incubated overnight in RPMI-1640 complete medium supplemented with 200 U/mL interleukin-3 (IL-3) (eBioscience), and subsequently cultured with 200 IU/mL interleukin-4 (IL-4) and 200 U/mL GM-CSF (eBioscience) for 6 days. Before harvesting, DCs were matured by incubation with 100 IU/mL TNFα (eBioscience) for another 16–20 h. To measure T-cell proliferation, isolated CD4+ T-cells were labeled with CFSE (eBioscience, catalog #65-0850-84) according to the manufacturer's procedure. Labeled T-cells ($1\times10^5$) and allogeneic DCs ($1\times10^4$) were cocultured with or without dose titrations of HLX10 in RPMI-1640 complete medium at 37˚C with an atmosphere of 5% $CO_2$. After 5 days incubation, CD4+ T-cell proliferation was measured by Cytomics FC500 Flow Cytometer (Beckman Coulter) and IL-2 secretion in culture supernatants was quantified by ELISA (BioLegend # 431801).

## *In vivo* efficacy studies

HT-29/hPBMC and NCI-H292/hPBMC xenograft NOD/SCID mouse model: A total of $5\times10^6$ HT-29 cells or NCI-H292 cells in 100 μL of DPBS were mixed with or without $1.7\times10^6$ hPBMCs in 100 μL of DPBS (ratio 3:1), and subcutaneously implanted into both side flanks of female NOD/SCID mice (6–8 weeks) (BioLASCO Taiwan). The next day post tumor

inoculation, the tumor-bearing mice were randomized into distinct groups. Mice were administered intraperitoneally with either antibodies or vehicle controls twice per week until the end of studies. Antibodies were prepared in the vehicle buffer (saline solution). The dose volumes were 10 mL/kg among studies. Tumors were observed and measured twice a week. Tumor volume was defined as TV (tumor volume) = (length × width$^2$)/2. The antitumor activities of treatment were expressed by tumor growth inhibition rate (TGI%). The formula is TGI (%) = [1-Mean (T$_{final}$-T$_{initial}$)/Mean(C$_{final}$-C$_{initial}$]×100%, where T is the average tumor volume of the treatment group and C is the average tumor volume of the control group. Tumor volume data were presented as mean ± SEM. Differences between the average tumor volume of control and treatment groups were analyzed using Student's T test. Statistical significance was determined at the level of $p < 0.05$ ($^*p \leq 0.05$, $^{**}p \leq 0.01$, $^{***}p \leq 0.001$, $^{****}p \leq 0.0001$).

EMT-6 syngeneic model: HuGEMM human PD1 knock-in mice (BALB/c nude background, from Gempharmatech Co,Ltd.) and B-hPD-1 (Biocytogen, Beijing, 101111, China) were used to inoculate EMT-6 tumor cells subcutaneously inoculated in the right flank with $0.5 \times 10^6$ cells in 0.1 mL PBS. Treatment was initiated 8 days later when tumors reached 95–100 mm$^3$. The mice were randomized into groups (N = 10) so that the average tumor sizes of all groups were similar, and treatment by intraperitoneal injections was initiated on day 8. Group 1 (vehicle group) received saline solution (10 μl/g) twice weekly; group 2 received 315 μg of HLX10 antibody twice weekly. Body weights were measured twice weekly to monitor toxicity. Tumor volumes were measured twice per week in two dimensions using a caliper, and the volume were calculated in mm$^3$ using the formula: "V = (L x W x W)/2, where V is tumor volume, L is tumor length (the longest tumor dimension) and W is tumor width (the longest tumor dimension perpendicular to L). Dosing as well as tumor and body weight measurements were conducted in a Laminar Flow Cabinet. The body weights and tumor volumes were measured by StudyDirector$^{TM}$ software (version 3.1.399.19). Tumor growth inhibition (TGI) is an indication of antitumor activity and calculated as: TGI (%) = 100 x (1-T/C). T and C are the mean tumor volume (or weight) of the treated and control groups, respectively, on a given day.

MDA-MD-231-HM TNBC cell line in humanized NSG mice: hu-CD34 NSG™ mice (strain background: PrkdcscidIl2rgtm1Wjl/SzJ) engrafted with human CD34$^+$ cells and >25% human CD4+ cells in the peripheral blood 12 weeks or later post engraftment were used for this study. Thirty hu-CD34 NSG$^{TM}$ mice were implanted orthotopically with the TNBC cell line MDA-MD-231($5 \times 10^6$ cells mixed with Matrigel® per mouse). Cells were injected into the mammary fat pad. Body weight and clinical observations were recorded once a week. Digital caliper measurements were initiated to determine tumor volume twice weekly, once tumors become palpable. Mice were randomized based on tumor volume when tumor volumes reached approximately 60–150 mm$^3$ (n = 8 mice per group). One group of mice were intraperitoneally dosed with 10 mg/kg of HLX10 on Study Day 0 and 5 mg/kg of HLX10 on Study Day 7, 14, 21, and 28. Another group of mice were intraperitoneally treated with Pembrolizumab (Keytruda®, provided by Jackson Laboratory, Sacramento CA) at a dose level of 10 mg/kg on study day 0 and 5 mg/kg on study Day 5, 10, 15, 20. The mice in vehicle group were intraperitoneally injected with PBS for a Q7D × 5 schedule.

All *in vivo* experiments were performed following an approved IACUC protocol in compliance with institutional guidelines. Animals were assessed daily for clinical symptoms and adverse effects (general sickness, respiratory distress, and impaired motility). For ethical reasons, animals were removed from the study if the tumor volume exceeded 2500 mm$^3$.

## HLX10 pharmacokinetic (PK) in cynomolgus monkeys

Dose-range finding of HLX10 were performed at JOININ Laboratories (Beijing Economic-Technological Development) testing facility in China. The studies were conducted in accordance with the protocol and applicable Covance Standard Operating Procedures (SOPs). All study-specific procedures were following the Animal Welfare Act Regulations (S1 File). Eighteen cynomolgus monkeys ($n$ = 6 for each group, 3 male and 3 female) were intravenous drip-treated with three single ascending doses of HLX10 (3, 10 and 30 mg/kg) on day 1 of the study. The infusion rate was 30 mL/kg/h with a dose volume of 10 mL/kg and the duration of infusion was 20 min. Blood samples of approximately 1 mL without anti-coagulant for pharmacokinetic analysis were collected from a forelimb or hindlimb subcutaneous vein (different from infusion site) of each monkey at designated time points before and after dosing until Day 29 (a total of 16 time points: pre-dose, immediate completion of infusion (set as 0 h), and 1, 2, 6, 24, 48, 72, 96, 120, 168, 240, 336, 408, 504 and 672 hours after beginning infusion). Serum concentrations of HLX10 were determined using a validated ELISA method, with LLOQ of 0.9 μg/mL. The method utilized human PD-1 protein as the capture reagent and a goat anti-human IgG horse radish peroxidase (HRP) conjugate as the detection antibody. Non-compartmental analysis was applied to the individual serum HLX10 concentration data. Pharmacokinetic analysis was performed using WinNonlin 6.4 (Pharsight Corporation, Mountain View, CA).

To measure free-PD-1 and occupied-PD-1, cynomolgus monkey blood was collected from 2 monkeys in each group at 2 h, 168 h (Day 8), 336 h (Day 15), 504 h (Day 22) and 672 h (Day 29) post-dose. A competition method was used to detect free PD-1 on the surface of T-cells using biotinylated HLX10 and PE-streptavidin by flow cytometry analysis on a BD Accuri C5 Flow Cytometer. Each blood sample was examined in two conditions: (1) add saturated amount of HLX10 and stained with the detection antibodies mixture, (2) stained with detection mixture only. Samples underwent a red blood cell lysis step and subsequent washes with the wash buffer (PBS+1% FBS) and blocking before each treatment. Stained and blocked samples were performed in duplicate and FITC-CD3 was used for gating T-cells.

## Results

### Biochemical characterization of a high affinity anti-PD-1 mAb

A fully human hinge-stabilized IgG4 mAb, HLX10, was developed against human PD-1 receptor. To generate HLX10, we immunized mice with the recombinant human PD-1 protein and identified the murine 1G4 (15M1) mAb clone, which bound to human and cynomolgus PD-1 proteins and inhibited PD-L1 binding. As shown in Fig 1A and 1B, cIG4 (human Fc chimera), binds PD-1 and inhibits PD-L1 binding similarly to Nivolumab analogue (Nivo). Subsequently, we humanized cIG4 using standard CDR grafting method into human germline light chain variable region IGKV1-39*01 and human germline heavy chain variable region IGHV3-11*04, without altering affinity and specificity to obtain HLX10 (Fig 2). HLX10 cross-reacts with cynomolgus monkey PD-1, but not murine PD-1s as shown by ELISA using recombinant PD-1 extracellular domains (ECDs) from the human, cynomolgus monkey, mouse, and rat. To confirm HLX10 binding to membrane PD-1, we assessed its binding to PD-1 transfected CHO cells. HLX10 binds to cell-surface expressed PD-1 with a single digit nanomolar affinity (3.7 nM), which was equivalent to the affinity of Nivo (Fig 3A). Neither Nivo nor HLX10 bind parental CHO cells, suggesting specific binding to PD-1 receptor. The monomeric binding affinities to human and cynomolgus monkey PD-1 ECDs were further estimated by biolayer interferometry (BLI) assay. Both human and cynomolgus PD-1 bind HLX10 with single digit nanomolar affinities (2.42 ± 0.49 and 4.78 ± 0.16 nM, respectively). The higher binding affinity

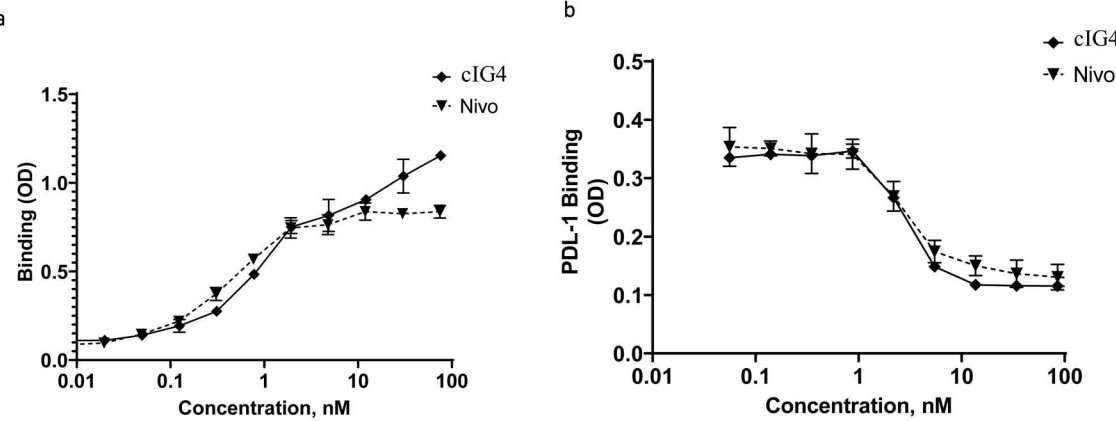

**Fig 1. Binding and PD-L1 ligand blocking of clG4 and Nivolumab antibodies.** ELISA assay comparing the binding of clG4 and Nivolumab reference to human PD-1 (a). ELISA assay comparing the ability of clG4 and Nivolumab reference to block binding of human PD-Ll to human PD-1 (b). Data points are means ± SD.

towards human PD-1 mainly resulted from the slower dissociation rate with human PD-1 than with monkey PD-1 [$k_d$ (s$^{-1}$): $3.29 \times 10^{-4}$ vs. $7.45 \times 10^{-4}$], whereas the association rate was relatively similar [$k_a$ (M$^{-1}$ s$^{-1}$): $1.93 \times 10^5$ vs $1.53 \times 10^5$]. We then compared HLX10's affinity to human PD-1 to other anti-PD-1 reference mAbs using the same BLI method. Nivolumab (Opdivo®), Pembrolizumab (Keytruda®), Sintilimab (Tyvyt®) and Toripalimab (JS001) were used for this comparison. As expected, these mAbs exhibited a range of binding affinities estimated by BLI (Table 1, S1 Fig). Sintilimab exhibited relatively higher affinity to human PD-1 compared to others, i.e., 2.09 nM vs. 2.42 nM for HLX10, 8.04 nM for Pembro, 11.9 nM for

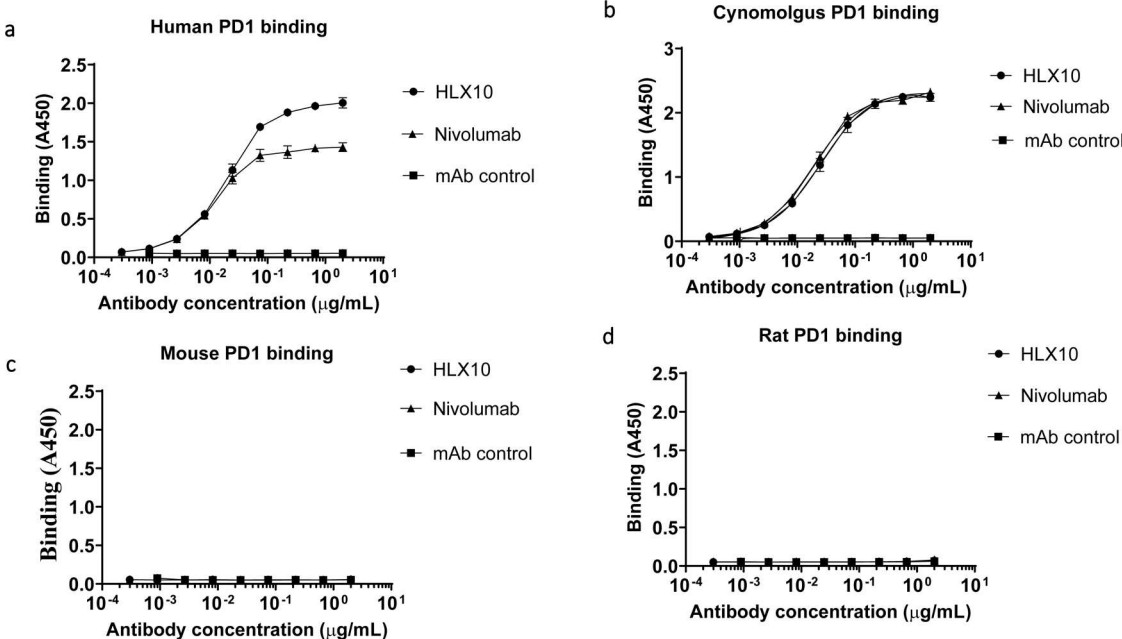

**Fig 2. ELISA binding of HLX10 to PD-1 ECD from different species.** The cross-reactivity of HLX10 binding to PD-1 extracellular domains (ECDs) of (a) human, (b) cynomolgus monkey, (c) mouse and (d) rat were determined using ELISA. HLX04, an unrelated human antibody mAb, was used as a negative control. All data points represent the mean man antibody human.

a

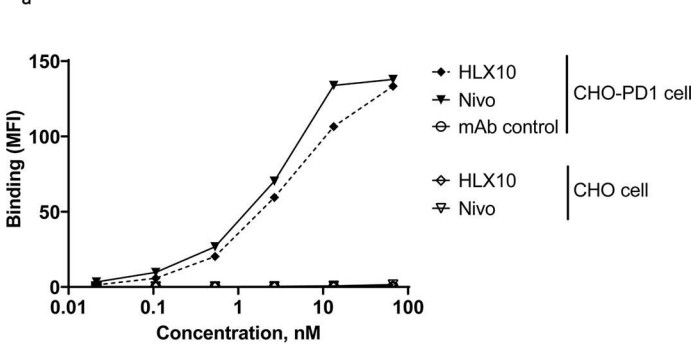

b

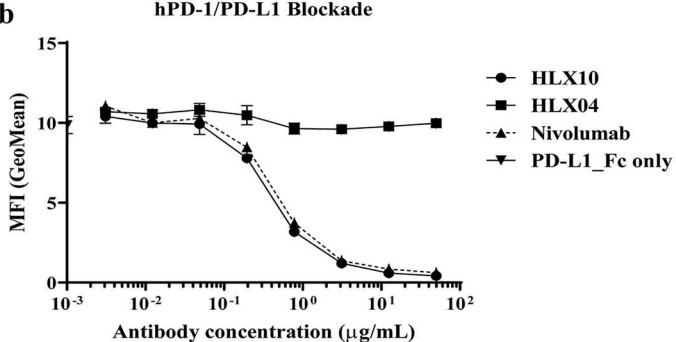

c

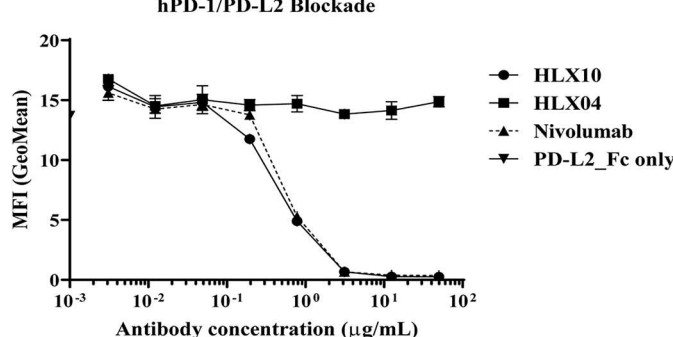

d

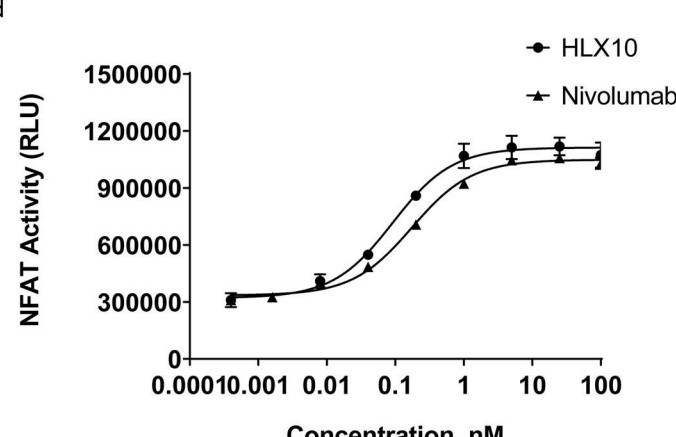

**Fig 3. Binding and PD-L1/2 ligand blocking of HLX10.** (a) The binding of HLX10 antibody to either CHO-S or PD-1 transfected CHO-S cells was assessed by flow cytometry. The reference anti PD-1 antibody (Nivolumab) and anti-PD-L1 were used as the positive control and negative control (mAb control), respectively. Binding is determined as the mean fluorescent intensity (MFI) of staining. (b) The binding of HLX10 to PHA activated human T-cells was tested by flow cytometry. The ability of HLX10 to inhibit either PD-L1 (c) or PD-L2 (d) binding to cell surface PD-1 was assessed using flow cytometry. Anti-VEGF humanized antibody was used as the negative control (mAb control). All data points represent the means ± SD of triplicate. (e) PD-L1 blocking reporter assay. PDL-1 blocking activity is presented as increase in luciferase signal upon blocking by either HLX10 or Nivolumab reference antibody. Each datapoint represents mean ± SD of duplicate (n = 2).

Nivo and 3.85 nM for Toripalimab. Nivo's affinity is slightly lower than the reporter affinity of Opdivo 3.06 nM (European Medicines Agency (EMA) [19]. A more recent publication reported the affinity for Nivo analogue to human PD-1 were 3.4 nM and 7.4 nM using different surface plasmon resonance (SPR) instruments [20]. These slight differences are expected and can be due to instrumentation and experimental conditions used. To confirm these data, we also included a SPR method. As shown in supplemental Fig 1B and 1C, HLX10 bound human PD-1 with ~ 10-fold higher affinity than Nivo; 0.32 nM vs. 3.05 nM, respectively. Therefore, both BLI and SPR data suggest that HLX10 affinity is better than Nivolumab analogue.

HLX10 was then evaluated for its ability to block PD-1/PD-L1 and PD-1/PD-L2 interactions using flow cytometry (FC). Representative curves of FC-determined blocking assays are shown in Fig 3B and 3C. The $IC_{50}$ of HLX10 were measured to be 4.36 nmol/L and 6.46 nmol/L for PD-1/PD-L1 and PD-1/PD-L2, respectively, while $IC_{50}$ of Nivo were 6.6 nmol/L and 12.44 nmol/L. The ability of HLX10 to inhibit PD1/PD-L1 signaling was further analyzed using a luciferase reporter system, where luciferase expression is under the control of a nuclear factor of activated T-cell (NFAT) promoter. As shown in Fig 3D, HLX10 shows dose dependent inhibition of NFAT activity. The estimated $EC_{50}$ was 0.09 nmol/L, while Nivo's $EC_{50}$ was 0.17 nmol/L. Therefore, HLX10 was able to specifically bind PD-1 receptor and efficiently block PD-L1 and PD-L2 signaling pathway with relatively higher $EC_{50}$ compared to Nivo analogue.

## Effect of HLX10 on immune cells

The effect of PD-1 blockade by HLX10 on the immune response was studied using the MLR in which CD4$^+$ T-cells recognize allogeneic monocyte-derived dendritic cells and results in T-cell proliferation and cytokine secretion. The secretion of IL-2 and IFNγ, was detected after co-culture of isolated T-cell with allogeneic monocyte-derived dendritic cells (DCs), in presence or absence of increasing concentrations of HLX10 for 2 to 5 days. In this assay, blockade of PD-1 by HLX10 potently induced IL-2 and IFNγ secretion in a dose-dependent manner, suggesting the T-cell activation and proliferation. Results from a representative DC/T-cell donor pair is shown in Fig 4A and 4B. To further assess effect on T-cell proliferation, CD4$^+$ T-cells derived

**Table 1. Kinetics and binding affinities of anti-PD-1 antibody to human PD-1 determined by BLI.**

| Sample | Human PD-1 | | |
|---|---|---|---|
| | $K_a$ [1/M s)] | $K_d$ [1/s] | $K_D$ [M] |
| HLX10 | $1.57 \times 10^5$ | $3.29 \times 10^{-4}$ | $2.42 \times 10^{-9}$ |
| Nivo | $1.28 \times 10^5$ | $1.50 \times 10^{-3}$ | $11.9 \times 10^{-9}$ |
| Pembro | $3.18 \times 10^5$ | $2.56 \times 10^{-3}$ | $8.04 \times 10^{-9}$ |
| Sintilimab | $1.44 \times 10^5$ | $3.00 \times 10^{-4}$ | $2.09 \times 10^{-9}$ |
| Toripalimab | $9.78 \times 10^4$ | $3.67 \times 10^{-4}$ | $3.85 \times 10^{-9}$ |

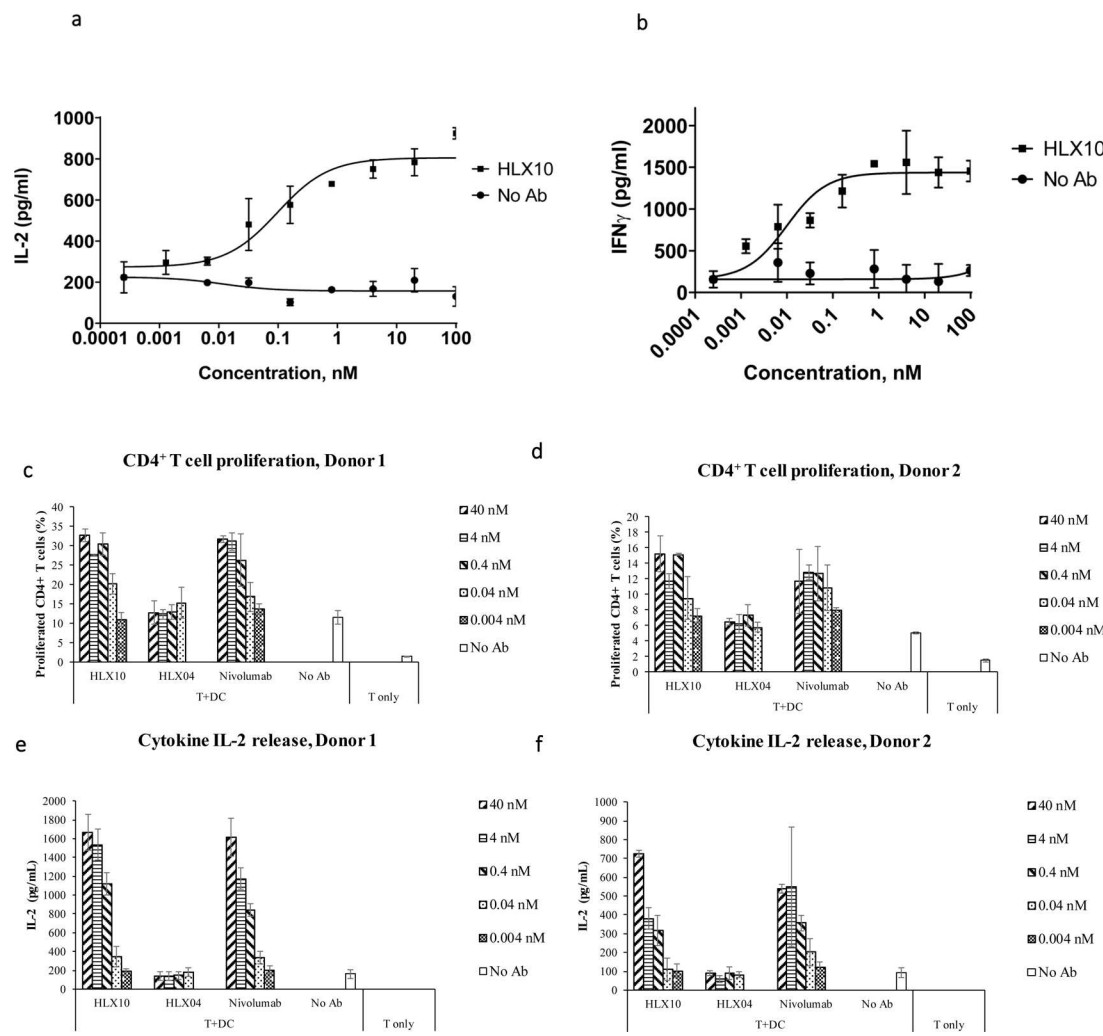

**Fig 4. IL-2 and IFN-γ dose–response curve of HLX10 in a mixed DC/CD4+ T-cell MLR assay.** (a, b) IL-2 and IFN-γ levels in the supernatants were determined by ELISA after 48 h and 5 days of culture, respectively. Data are presented as increase in cytokine levels relative to untreated cell (no Ab). Each datapoint represents mean ± SD (n = 3). (c, d) Effect of increasing doses of HLX10 and Nivolumab on CD4+ proliferation, as measured by CFSE staining by flow cytometry of two independent donor pairs. Data are presented (horizontal bar) as percent increase in CFSE stained population relative to untreated, no antibody and T-cell only controls. Each bar represents the mean ± SD. Nivolumab and HLX04 (anti-VEGF) were used as the positive control and negative controls, respectively. (e, f) Same MLR samples were assessed for IL-2 secretion by ELISA. Data are presented as increase in IL-2 levels relative to untreated, no antibody and T-cell only controls. Each graph represents a donor pair, and the horizontal bar is the mean ± SD.

from the MLR experiments were stained with carboxyfluorescein diacetate succinimidyl ester dye (CFSE), and analyzed for T-cell activation, proliferation, and cytokine release upon treatment with HLX10 and Nivo by flow cytometry. As shown in Fig 4C and 4D, HLX10 treatment results in a titratable T-cell proliferation enhancement in both donor pairs. At as low as 0.04 nmol/L, HLX10 started to show enhanced CD4+ T-cells proliferation, in a manner similar to Nivo, while the negative control antibody, HLX04, did not show any effect on T-cell proliferation. Similarly, HLX10 and Nivo augmented IL-2 in both donors (Fig 4E and 4F).

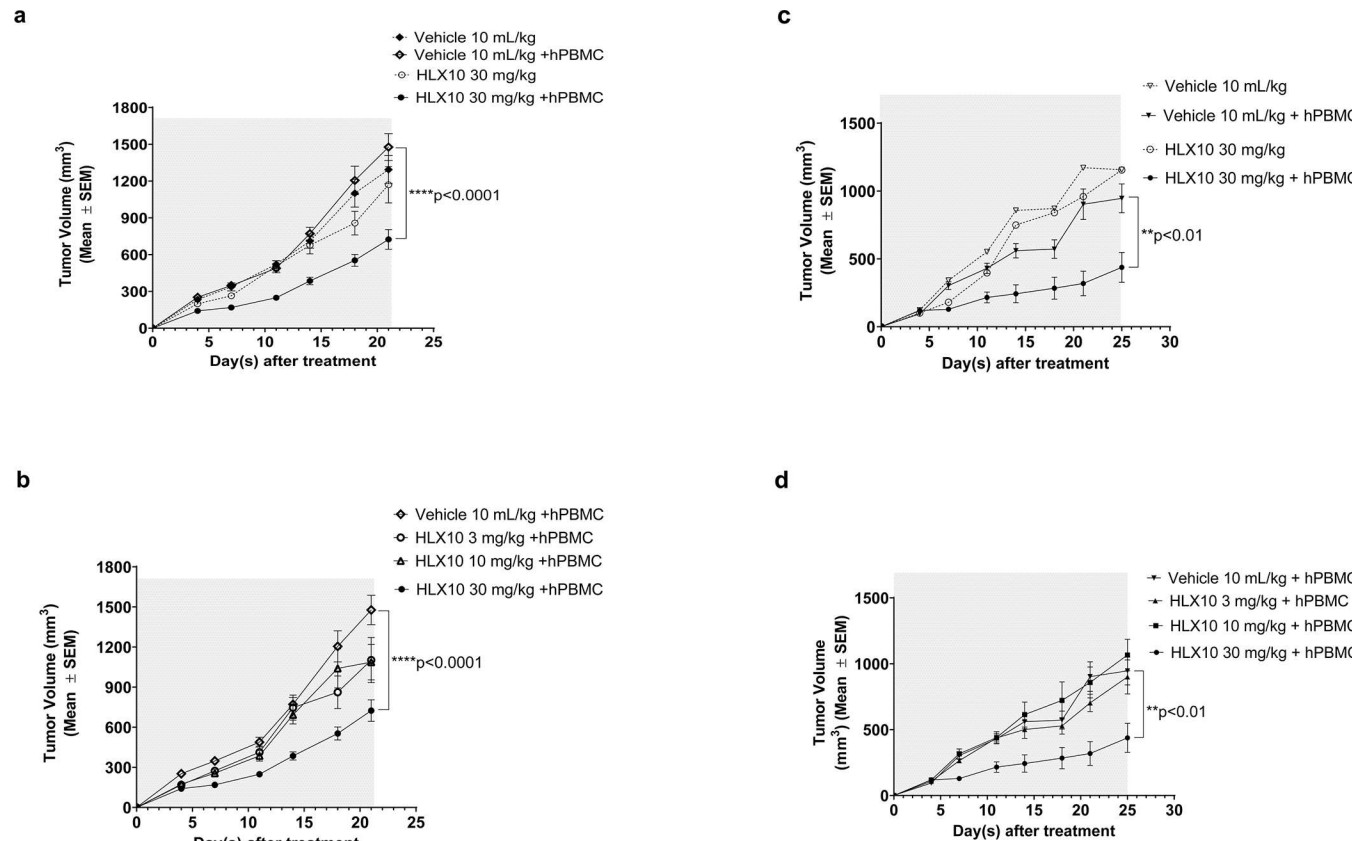

**Fig 5. Tumor growth inhibition of HLX10 in HT-29- and NCI-H292 -hPBMC co-mixture xenograft mice models.** The mice (n = 4/group) were engrafted subcutaneously with the mixture of human colon cancer cell lines HT29 (a, b), NCI-H292 (c, d), and freshly isolated human PBMC (cancer cells: PBMC = 2:1 and PBMC = 3:1, respectively). HLX10 (3, 10 and 30 mg/kg) antibody and vehicles were intraperitoneally injected into mice twice a week from day 1 as indicated (gray area). As controls, tumor cells only and HLX10 without hPBMC (30 mg/kg) were included. TGI are represented as the means ± SEM.

## *In vivo* efficacy studies of HLX10

The capability of HLX10 to activate T-cells and inhibit tumor growth was explored *in vivo* using various tumor models. HLX10 was initially tested in two tumor/hPBMC co-mixture cancer models, since HLX10 does not cross-react with murine PD-1. We used HT-29/hPBMC (colorectal cancer, BRAF$^{V600E}$) and NCI-H292/hPBMC (non-small-cell lung cancer) xenograft tumor in NOD/SCID. No significant difference in tumor growth is observed between the vehicle and vehicle + hPBMC control groups (Fig 5A and 5C). In HT-29 xenograft model, only 30 mg/kg of HLX10 plus hPBMCs significantly inhibited tumor growth ($p<0.0001$). The tumor growth inhibition rate (TGI%) was 51% (Fig 5A and 5B). Similarly, only HLX10 at 30 mg/kg plus hPBMCs significantly inhibited NCI-H292 tumor growth (TGI% = 54%) (Fig 5C and 5D). Whereas HLX10 treatment at 30 mg/kg did not result in an anti-tumor effect in immuno-deficient SCID mice without hPBMC, which lack T-cells but retain functional natural killer cells, demonstrating that a functioning hPBMC immune cells are required for HLX10 *in vivo* activity.

To confirm HLX10 *in vivo* activity, we next used EMT-6 mammary carcinoma cancer model inoculated in BALB/c human PD-1 knock-in transgenic mice. The EMT-6 is considered an anti-PD-1 resistant tumor, due to an immune excluded phenotype, macrophage-derived suppressive infiltration, immunosuppressive cytokine such as TGF-β, and epithelial-to-

a

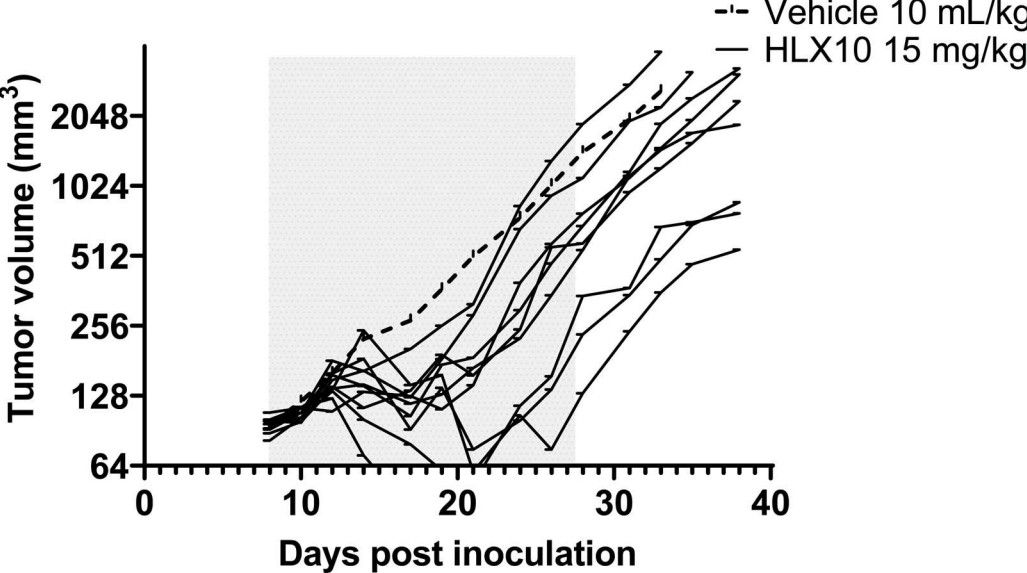

**Fig 6. Growth of EMT breast cancer syngeneic tumors was measured in PD-1 knock-in mice following 6 injections (gray area) of either vehicle buffer (10 mL/Kg),or HLX10 (15 mg/kg), and 2 weeks after the last dose (day 25); n = 10 mice/group.** Average tumor size of vehicle control (plotted as dashed line) and individual growth curves (solid lines) of HLX10 are shown.

mesenchymal transition (*EMT*) [21–23]. Upon treatment of established *EMT-6*/tumors in BALB/c mice, HLX10 was able to significantly inhibit tumor growth with some responding and nonresponding mice (Fig 6A), with an average TGI of 47.75% on day 33 (*p* = 0.0269 vs. vehicle group), suggesting that other suppressive mechanisms undermined the anti-tumor activity in these mice. One mouse in this group was completed cured. HLX10 *in vivo* activity was also demonstrated using the murine intestinal cancer MC38 model, which was established in C57BL/6 B-hPD-1 knock-in transgenic mice (S2 Fig).

### *In vivo* comparison of HLX10 to Nivolumab and Pembrolizumab

To further validate HLX10 *in vivo*, the antitumor effect of HLX10 was first compared to Nivo in NCI-H292/hPBMC co-mixture model, as described above, with hPBMCs from different donors. Likewise, there was no significant difference in tumor growth between tumor ± hPBMC (S3 Fig). Moreover, neither HLX10 nor Nivo inhibit NCI-H292 tumor growth in absence of hPBMC. However, in the presence of hPBMC (Fig 7A), HLX10 dosed at 3, 10 and 30 mg/kg showed a pronounced effect on tumor growth with TGIs of 58, 74 and 82% at day 25, respectively (*p*<0.05, *p*<0.001, and *p*<0.0001). In contrast, only the 30 mg/kg dose of Nivo yielded a significant tumor inhibition (TGI = 77%) at day 25. Notably, HLX10 was more effective using this new hPBMC donor compared to Fig 5D, likely due to more activated phenotype.

To further explore the anti-tumor HLX10 comparability to approved Pembrolizumab (Keytruda®), we used triple-negative breast cancer (TNBC) MDA-MD-231 model, which was orthotopically implanted in the humanized CD34+ (hu-CD34) NSG™ mice. As shown in Fig 7B, HLX10 achieved similar efficacy as Pembrolizumab. In summary, our monotherapy *in*

**a**

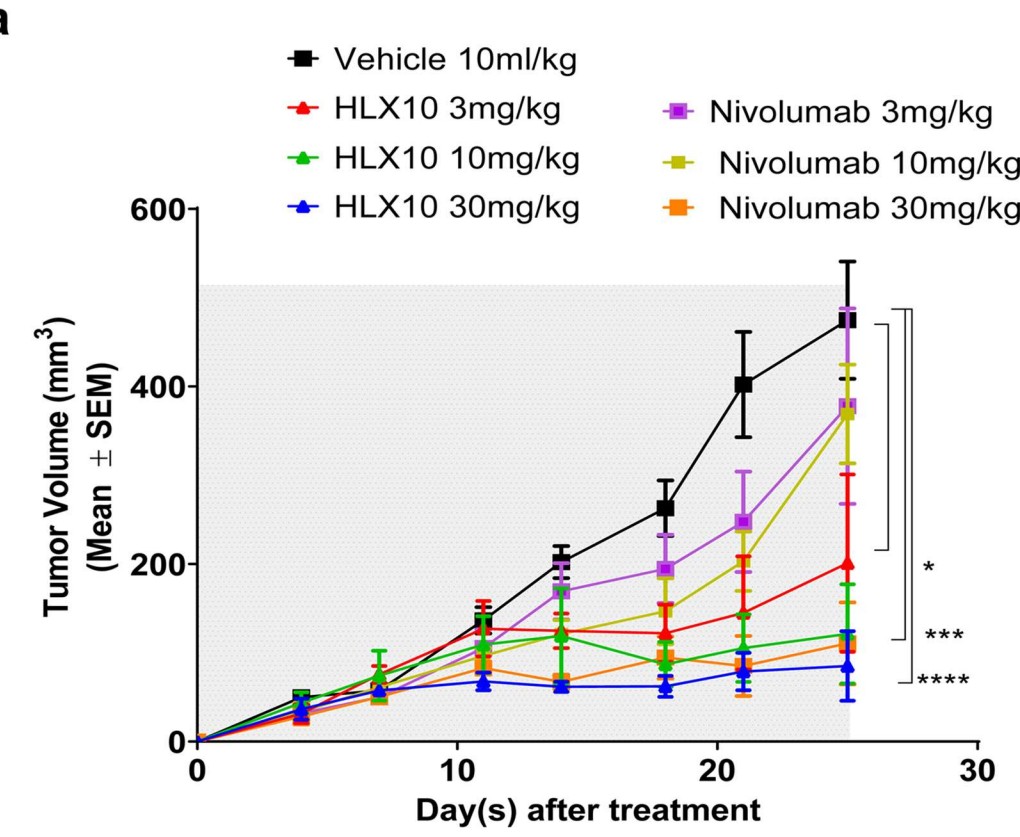

**b**

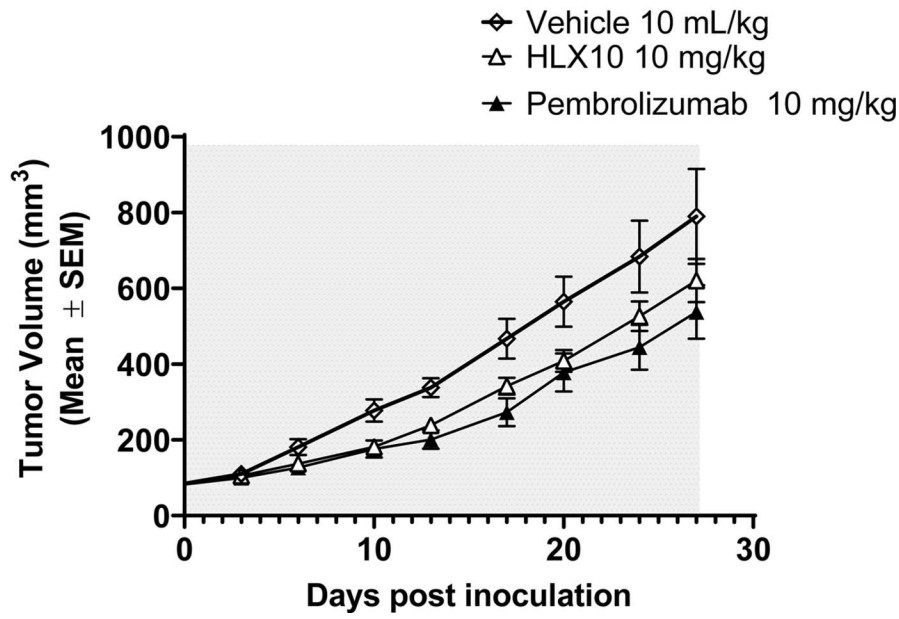

**Fig 7. HLX10 *in vivo* antitumor activity compared to that of Nivolumab and Pembrolizumab.** (a) NOD/SCID mice engrafted with NCI-H292/hPBMC co-mixture (same as Fig 5C and 5D), and treated with vehicle, HLX10 or Nivolumab at the indicated doses; n = 8 mice/group. (b) CD34+ humanized NSG mice engrafted with MDA-MB-231-HM model and treated with vehicle, HLX10 or pembrolizumab (Keytruda®), intraperitoneally. Antibodies were administered at a dose level of 10 mg/kg on day 0 and the remaining doses were maintained at 5 mg/kg for a Q7D × 5 schedule (gray area). The mice in vehicle group were intraperitoneally injected with PBS following the same Q7D × 5 schedule. TGI are represented as the means ± SEM.

*vivo* studies demonstrated that HLX10 was able to generate significant anti-tumor efficacy against various cancer types, providing a rationale for clinical evaluation in cancer immunotherapy.

## HLX10 synergy with Bevacizumab, an anti-VEGF blocking antibody

In recent years, studies have found that many cancers are closely related to abnormal blood vessel formations, and normalization of the tumor vasculature is thought to promote and synergize with immunotherapy [24]. To examine HLX10's ability to synergize with anti-VEGF blockade, we evaluated *in vivo* activity of HLX10 in combination with Bevacizumab (HLX04, Avastin biosimilar) in the human colorectal cancer cell line HT-29/hPBMC xenograft model. As shown in Fig 8A, HLX10 and Bevacizumab monotherapy groups showed a significant tumor growth with a TGI% of 35% and 52%, respectively ($p<0.05$). Compared to the monotherapy groups, the combination of HLX10 and Bevacizumab resulted in a stronger inhibition of tumor growth with TGI% of 67% ($p<0.01$). Although the HLX10 plus Bevacizumab combination group appeared to have better anti-tumor activity than the single-agent treatment group, there was no statistical difference between the mono and combination therapy groups ($p = 0.1$). To confirm the synergy between these two mechanisms, we also examined the *in vivo* anti-tumor additive effect of HLX10 plus Bevacizumab in the non-small-cell lung cancer, NCI-H292/hPBMC xenograft model. As demonstrated in Fig 8B, the combination of HLX10 plus Bevacizumab significantly inhibits tumor growth, much stronger than either HLX10 or Bevacizumab single agents. Contrary to HLX10, which did not inhibit tumor growth, Bevacizumab was highly effective at 5 mg/kg ($p<0.01$). The TGIs% at days 21 were 92% vs. 65% ($p<0.001$), for the combination and the Bevacizumab monotherapy groups, respectively. This data suggests that HLX10 and Bevacizumab could mutually enhance each other effect through several mechanism, including vessel normalization, TIL infiltration and restoration of immune active microenvironment [25]. Thus, the combined use of an anti-VEGF antibody Bevacizumab, and anti-PD-1 antibody HLX10 may provide benefit for cancer patient, in particular lung and colon cancer patients.

## Overall structure of PD-1 in complex with HLX10 Fab

The crystal structure of human PD-1 ectodomain domain (residues 32–160, S93C) in complex with HLX10 Fab fragment was determined and refined to a resolution of 1.78 Å, with $R_{work}$ and $R_{free}$ values of 0.16 and 0.20, respectively (S1 File and Table 2). The space group of this protein complex crystal is P1, with one hPD-1/HLX10-Fab complex molecule in each asymmetric unit. The structure shows that HLX10 Fab adopts a typical β-sandwich immunoglobulin fold structure with the conventional intra-molecular disulfide bonds (LC: Cys23-Cys88, Cys134-Cys194; HC: Cys22-Cys96, Cys143-Cys199) (Fig 9A). The complex formation buries an interaction surface area of 1042.8 Å$^2$ on Fab fragment of HLX10, which is similar with the common value observed in other antigen/antibody complexes. All six-complementarity determining region (CDR) loops are well-defined by electron density map in HLX10 Fab structure except K24 (LCDR2) and K65 (HCDR2), which are not contact residues. All six CDRs of VH

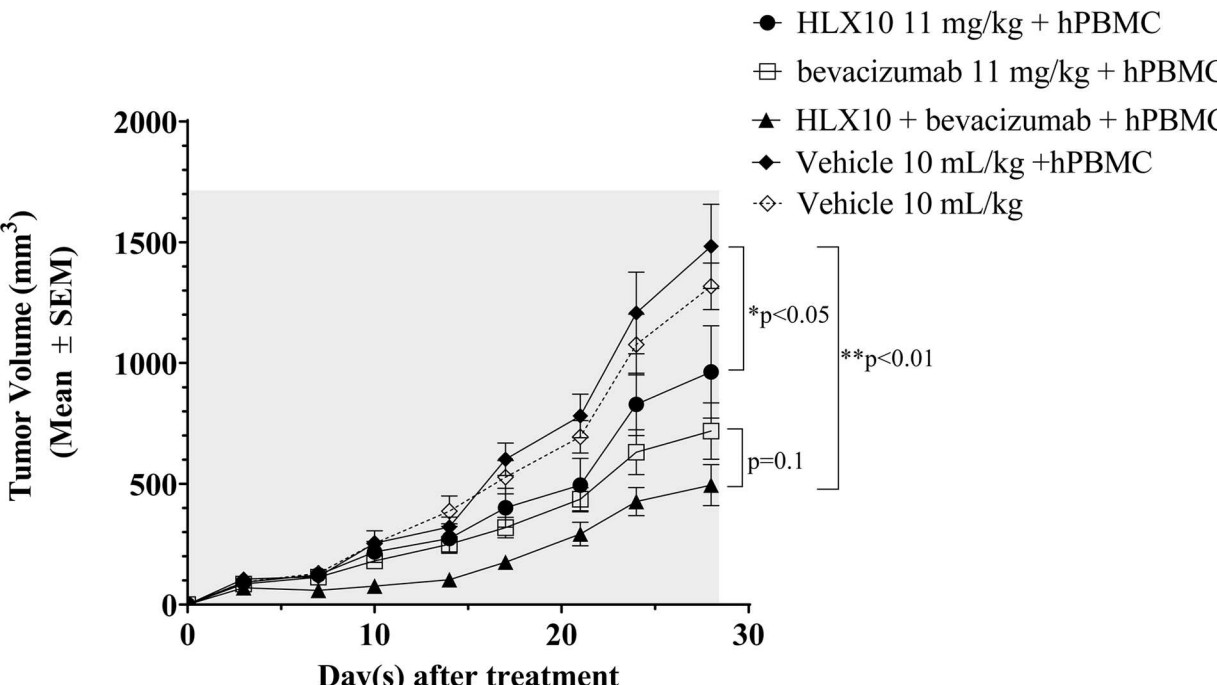

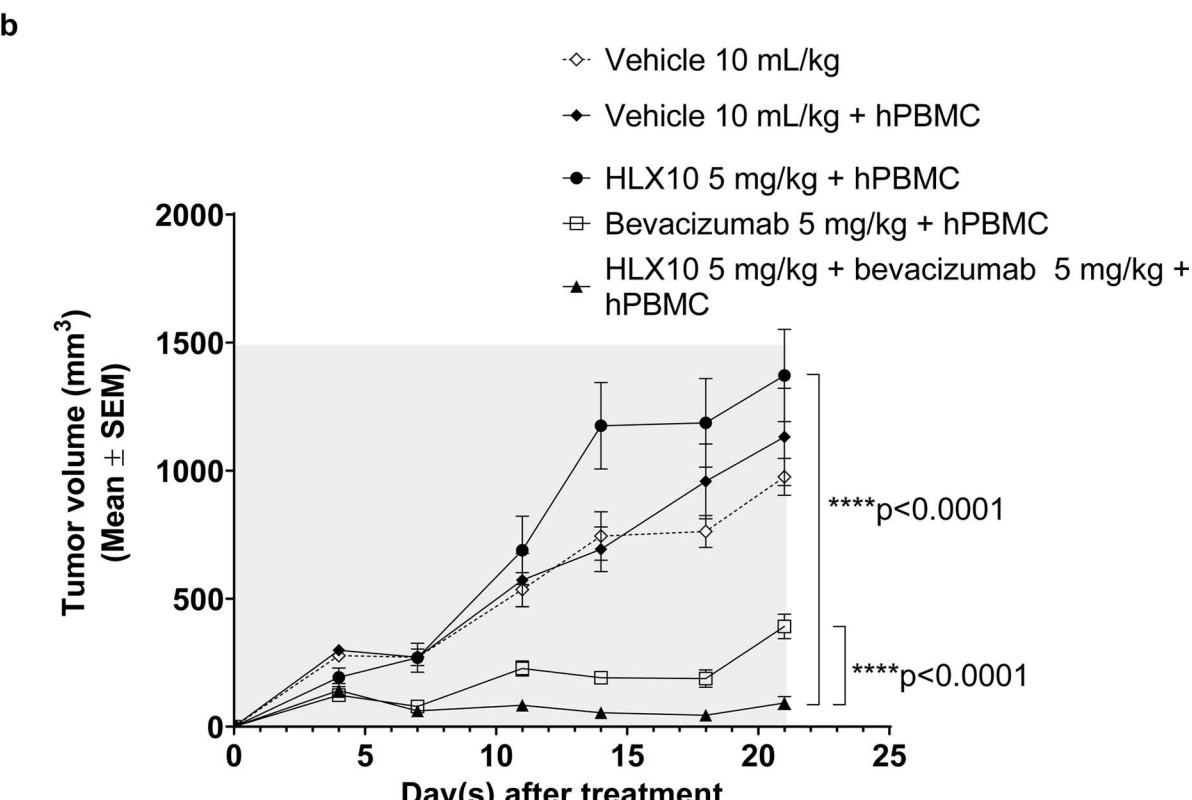

**Fig 8. Anti-tumor activity of HLX10 and Bevacizumab in HT-29 and NCI-H292 hPBMC co-mixture xenograft models.** NOD/SCID mice engrafted with either HT-29/hPBMC or NCI-H292/hPBMC co-mixture models and treated with vehicle, HLX10, Bevacizumab or HLX10 + Bevacizumab combination at the indicated doses; n = 8 mice/group. Antibodies and vehicle controls were administered intraperitoneally twice per week until the end of the study. TGI are the means ± SEM.

and VL of HLX10 Fab contributed to interaction with PD-1 (Fig 9B). however, LCDR1 have the least contact with hPD-1, where Thr30 forms a hydrogen bond with backbone oxygen of Ala132 of hPD-1 through a water molecule. The binding interface is formed by residues in three loop regions in PD-1, such as C'D loop, BC loop and the FG loop. Most prominently, Arg86 of hPD-1 which is in C'D loop provide a key part of the interaction. It forms two salt bridge with Asp104, two hydrogen bonds with Ser98, one hydrogen bond with Tyr99 of HCDR3, while it also forms a π-π interaction with Tyr32 from HCDR1 (Fig 9C and 9D, and S4 Fig).

To characterize HLX10 mechanism of PDL-1 blocking, we compared hPD-1/HLX10-Fab structure to crystal structures of the PD-1 receptor in complex with hPD-L1 (PDB: 4ZQK) and

**Table 2. Data collection and refinement statistics.**

| **Data collection** | |
|---|---|
| Space Group | P1 |
| Cell dimensions | |
| a, b, c (Å) | 38.985 45.014 80.89 |
| α, β, γ, (°) | 93.60, 102.16, 97.34 |
| Resolution (Å) | 30.93–1.78 (1.84–1.78) |
| $R_{merge}$ (%) | 0.077 (0.50) |
| I / σI | 28.9 (2.75) |
| Completeness (%) | 94.63 (92.97) |
| Redundancy | 7.1 (6.8) |
| **Refinement** | |
| No. reflections | 48157 (4735) |
| $R_{work}$ / $R_{free}$ (%) | 0.169/0.204 |
| No. atoms | |
| Protein | 4100 |
| Water | 481 |
| Average B-factor | 32.23 |
| Protein | 31.32 |
| Water | 40.06 |
| R.m.s. deviations | |
| Bond lengths (Å) | 0.016 |
| Bond angles (°) | 1.35 |
| Ramachandran plot statistics (%) | |
| Most favoured | 97.52 |
| Allowed | 2.48 |
| Disallowed | 0.0 |

*Values in parentheses are for highest-resolution shell.

Values in parentheses are for the highest resolution shell. $Rmerge = \Sigma h\Sigma i|Ih,i-Ih|/\Sigma h\Sigma iIh,i$, where $Ih$ is the mean intensity of the $i$ observations of symmetry related reflections of $h$. $R = \Sigma|Fobs-Fcalc|/\Sigma Fobs$, where $Fcalc$ is the calculated protein structure factor from the atomic model (Rfree was calculated with 5% of the reflections selected randomly).

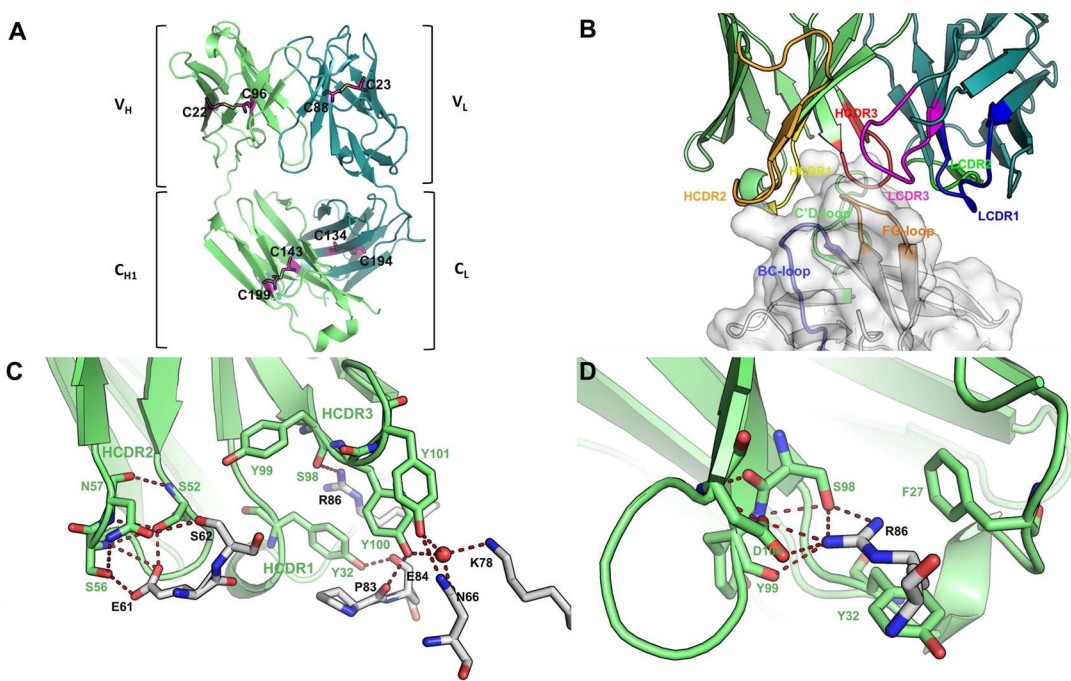

**Fig 9. Interaction between HLX10 and hPD-1.** (a) HLX10 Fab adopts a typical immunoglobulin fold structure. Heavy chain and light chain are colored in lime and deep teal, respectively. Intra-molecular disulfide bonds are shown in magenta sticks. (b) Side view of binding interface between HLX10 and hPD-1. hPD-1 is represented as surface in white, with BC loop, C'D loop and FG loop in light orange, light blue and light green, respectively. HCDR1, HCDR2 and HCDR3 loops are colored in yellow, orange, and red, respectively, while LCDR1, LCDR2 and LCDR3 loops are colored in blue, green, and magenta, respectively. (c, d) Interaction within interface of CDRs loop of heavy chain and hPD-1. Arg86 of hPD-1 is involved in several hydrogen bonds and salt bridges with HLX10.

hPDL-2 (PDB: 6UMT). These structures showed that the interface buries 779Å$^2$ and 893.5Å$^2$ of the solvent-accessible surface area on PD-1 ECD, for PDL-1 and PDL-2, respectively (Fig 10A and 10B). It is reported that the intermolecular interaction between PD-1 and PD-L1 occurs mainly via the *CC'FG sheet* and induces the PD-1 CC' loop to change in conformation slightly, enabling it to close around the PD-L1 molecule on binding. Notably, our hPD-1/ HLX10-Fab structure indicates that the epitope recognized by HLX10 overlaps largely with the residues in PD-1 responsible for the interaction with PD-L1 and PDL-2 (Fig 10C). Six residues (Asn66, Lys78, Ile126, Lys131, Ala132, and Ile134) of PD-1 participate in hydrogen bonds and hydrophobic interactions with both HLX10 Fab and PDL-1. Since the interaction of HLX10 Fab with PD-1 is heavily dependent on the flexible C'D loop of PD-1, which is not directly involved in the interaction with PD-L1, this interaction ensures that HLX10 competes with the binding of PD-L1. Moreover, comparison of unbound hPD-1 structure (PDB: 3RRQ) with hPD-1/HLX10-Fab suggests that HLX10 binding HLX10 induces small local conformational change in CC' loop of hPD-1 (Fig 10D), which could be incompatible with PD-L1 binding.

## Comparison to Pembro and Nivo structures

To further elucidate the mechanism of HLX10-Fab blocking ability, we superposed the PD-1-HLX10-Fab complex structure with PD-1-nivolumab-Fab (PDB:5WT9) and PD-1- Pembro-Fab (PDB: 5GGS). Fig 11A and 11B show that there is a significant overlap between the epitopes of HLX10 and those of Nivo and Pembro with similar structure and orientation. Notably,

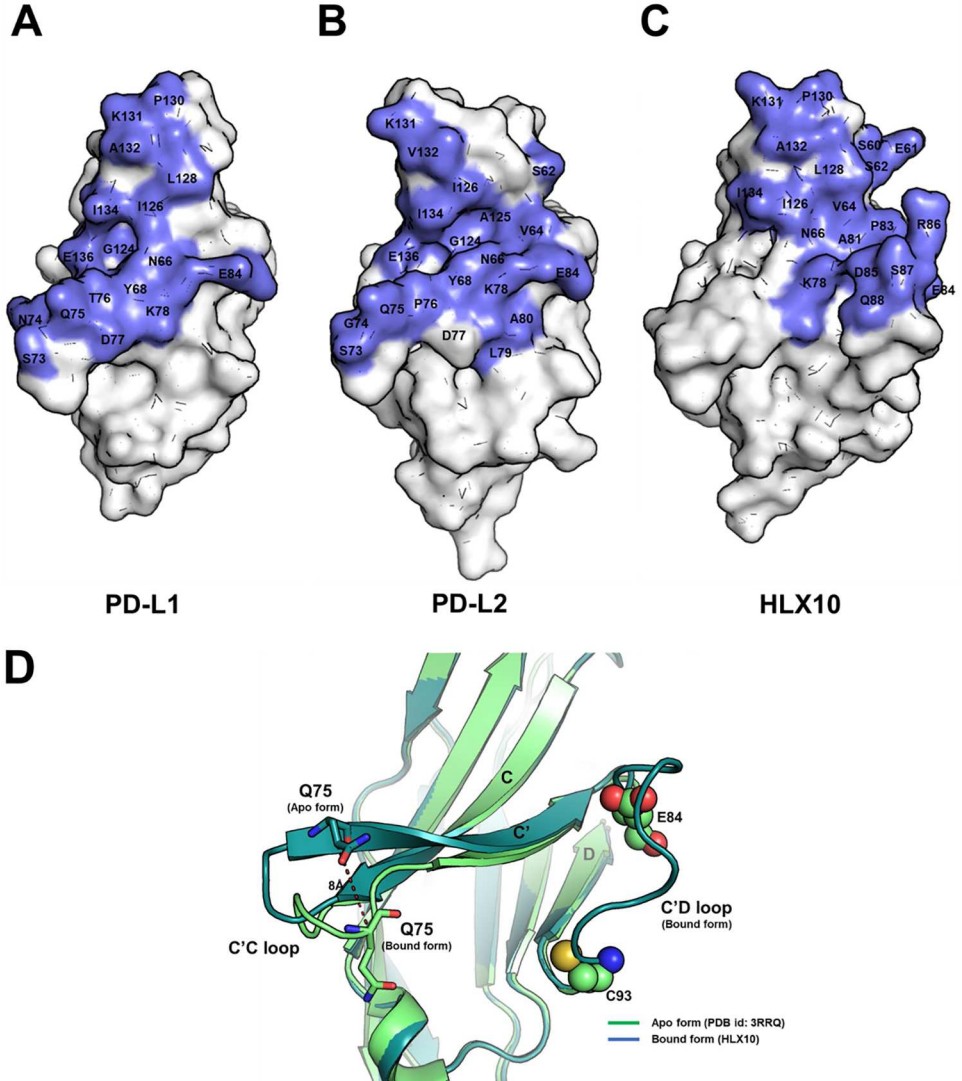

**Fig 10. Binding epitope of HLX10 overlaps with PD-L1 and PD-L2.** (a-c) Comparison of binding interface of PD-L1, PD-L2 and HLX10. hPD-1 is represented as white surface with binding interface of binders (PD-L1, PD-L2 and HLX10) colored in slate. The C'D loop of hPD-1 is missing in hPD-1/hPD-L1 (PDB: 4ZQK) and hPD-1/hPD-L2 structures (PDB: 6UMT) because of structure flexibility. (d) rearrangement of CC' loop of hPD-1 for HLX10 binding. Cα atom of Gln75 displays an 8 Å movement. Except C'C loop of hPD-1, there is no significant conformational change between apo and bound form of hPD-1 structures (overall backbone RMSD = 0.351Å). The C'D loop is missing (D85-D92) in apo form hPD-1 structure (PDB: 3RRQ) because of structure flexibility.

HLX10 shows an opposite heavy chain (HC) and light chain (LC) usage compared to Pembro. As previously described, Nivo binds PD-1 by using the N-terminal extension, FG and BC loops as a mode for binding, while the overlapping binding surface shared by the VH region of HLX10 and PD-L1 resides mostly on the FG and C'Ds loops. Furthermore, since our construct design of hPD-1 (W32-P160), does not contain the N-terminal loop (L25-R30), it suggests that the N terminal loop is not essential for HLX10 binding. In contrast, the interaction of Pembro with hPD-1 is heavily dependent on the flexible C'D loop of PD-1, demonstrating that HLX10 epitope is more similar to pembro than Nivo. As summarized in Table 3, HLX10 binds shared several residues with Pembro and Nivo. Furthermore, HLX10's epitope region shows a much

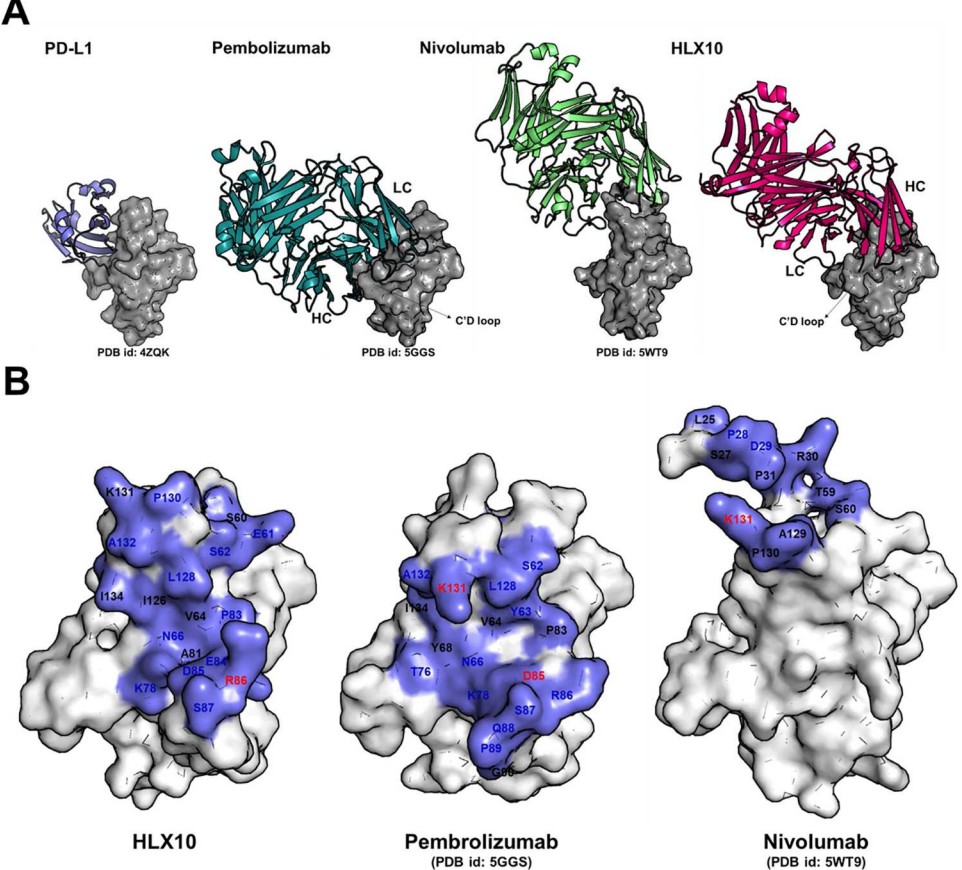

**Fig 11. Comparison of binding epitope of HLX10 with Pembrolizumab and Nivolumab.** (a) Binding of Nivolumab is mainly located on the top (N-terminal extension, BC loop, and FG loop) of h-PD-1, whereas Pembrolizumab and HLX10 is located on CC'FG sheets of h-PD-1. hPD-1 is represented as surface in grey. PD-L1, Pembrolizumab, Nivolumab, and HLX10 are colored in slate, deep teal, lime and magenta ribbon, respectively (b) Residues of hPD-1 contribute to the interactions with different binders. hPD-1 is represented as white surface with binding epitope colored in slate. The residues which are involved in hydrogen bond, salt bridge and hydrophobic interaction are colored in blue, red and black, respectively.

greater overlap with Pemro and PD-L1 binding site than the epitope region of Nivo. HLX10 interface occupies a solvent-accessible overlapping surface area of 445Å$^2$ (55% of PD-L1 surface) on PD-1 vs. 27% and 60% for Nivo and Pembro, respectively.

In summary these structures analysis suggests that HLX10, Nivo and Pembrro exhibit similar mechanism of action whereby they competitively inhibit PD-L1 binding by direct occupancy and steric blockade of the PD-L1 binding site.

## Monkey single dose pharmacokinetics

As shown in Fig 12, HLX10 exhibits a dose-proportional pharmacokinetics and the exposure (measured by $C_{max}$ and $AUC_{last}$) was dose-proportional with $C_{max}$ values increasing 3.3 and 9.9-fold and $AUC_{last}$ increasing 3.8 and 12.1-fold at 10 and 30 mg/kg (compared to values at 3 mg/kg) (Table 4). Mean Residence Time (MRT) ranged from 153.02 to 231.28 hours, $T_{1/2}$ ranged from 137.97 to 256.99 hours. To determine receptor occupancy (RO) after HLX10 administration, we measured free PD-1 using a competition assay. Fig 12B shows that animals in all three groups maintained >90% blockade of free-PD-1 upon HLX10 administration.

**Table 3. Comparison of Paratope and Epitope of HLX10, Pembrolizumab and Nivolumab.**

| PD1 | HLX10 | PD1 | Pembrolizumab | PD1 | Nivolumab |
|---|---|---|---|---|---|
| **C'D loop**<br>P83, E84, D85, R86, S87 | **HCDR1**<br>N31, F32<br>**HCDR2**<br>S52, G54, S56, N57<br>**HCDR3**<br>S98, Y100, Y101, D104, F105<br>**LCRD2**<br>H55, T56<br>**H-Framework**<br>F27 | **C'D loop**<br>P83, D85, R86, S87, Q88, P89, G90 | **HCDR1**<br>Y35<br>**HCDR2**<br>N59, G50, I51, N52, G57, T58<br>**HCDR3**<br>R99<br>**LCDR1**<br>Y36<br>**LCDR3**<br>R96 | **N-terminal loop**<br>S25, S27, P28, D29, R30 | **HCDR1**<br>N31, S32, G33<br>**HCDR2**<br>W52, Y53, K57 |
| **FG loop**<br>L128, P130, K131, A132, I134 | **HCDR2**<br>Y59<br>**LCDR1**<br>T30<br>**LCDR3**<br>H91, Y92, T93, I94, W96 | **FG loop**<br>L128, K131, A132, I134 | **HCDR3**<br>R102, M105<br>**LCDR2**<br>Y53, L54,<br>**L-Framework**<br>E59 | **FG loop**<br>A129, P130, K131 | **HCDR3**<br>N99, D100, D101, Y102<br>**LCDR2**<br>N53, R54, A55, T56<br>**L-Framework**<br>L46, L47, I48, Y49 |
| **BC loop**<br>S60, E61, S62 | **HCDR2**<br>S52, G54, S56, N57<br>**HCDR3**<br>Y99 | **BC loop**<br>S62, F63 | **LCDR1**<br>Y34<br>**LCDR2**<br>Y57 | **BC loop**<br>T59, S60, E61 | **HCDR1**<br>I27, T28, N31 |
| **CC'FG sheets**<br>V64 (C), N66 (C) K78(C'), I126(F) | **HCDR3**<br>Y101, D104, F105<br>**LCRD2**<br>W50 | **CC'FG sheets**<br>V64 (C), N66 (C), Y68 (C), T76 (C'), K78(C') | **HCDR1**<br>Y33<br>**HCDR3**<br>Y101, R102, F103 | | |

Moreover, there was no significant reduction in RO as the plasma concentration decreased during the 4 weeks of dosing. HLX10 was well tolerated in this study as well as subsequent repeated dose studies with 4 weekly dosing for up to 100 mg/kg.

## Receptor occupancy measurement of HLX10

To confirm HLX10 specific binding to its cellular target in T-cells and elucidate pharmacodynamic (PD) biomarker in human, we measured receptor occupancy (RO) *in vitro*. Serially diluted HLX10 was added into human whole blood samples from four healthy donors to mimic serum concentrations of HLX10 after treatment. RO on peripheral blood CD3$^+$ T-cells was detected with a competing antibody and measured using flow cytometry. As illustrated in Fig 13, the target occupancy of HLX10 on CD3$^+$ T-cells from all subjects increased in a concentration dependent manner. The mean concentration of all blood samples required to achieve 50% RO was 0.47 μg/mL. Greater than 70% RO was reached at 2 μg/mL of HLX10 in all donor blood samples. These *in vitro* data support the dose selection for human clinical evaluation.

## Discussion

PD-1 receptor has become one of the pivotal immune checkpoint molecules for cancer therapy. Antibodies blocking PD-1 and its ligands, as monotherapy or in combination with other agents, have shown good efficacy in a broader spectrum of cancer types. However, only a fraction of patients develops durable clinical responses, while most patients develop only transient responses or do not respond at all. To improve overall efficacy, considerable effort is currently explored, including novel anti-PD-1/PD-L1 combinations and bispecific antibodies. Although several mAbs have been approved for cancer therapy by FDA and NMPA, screening of new

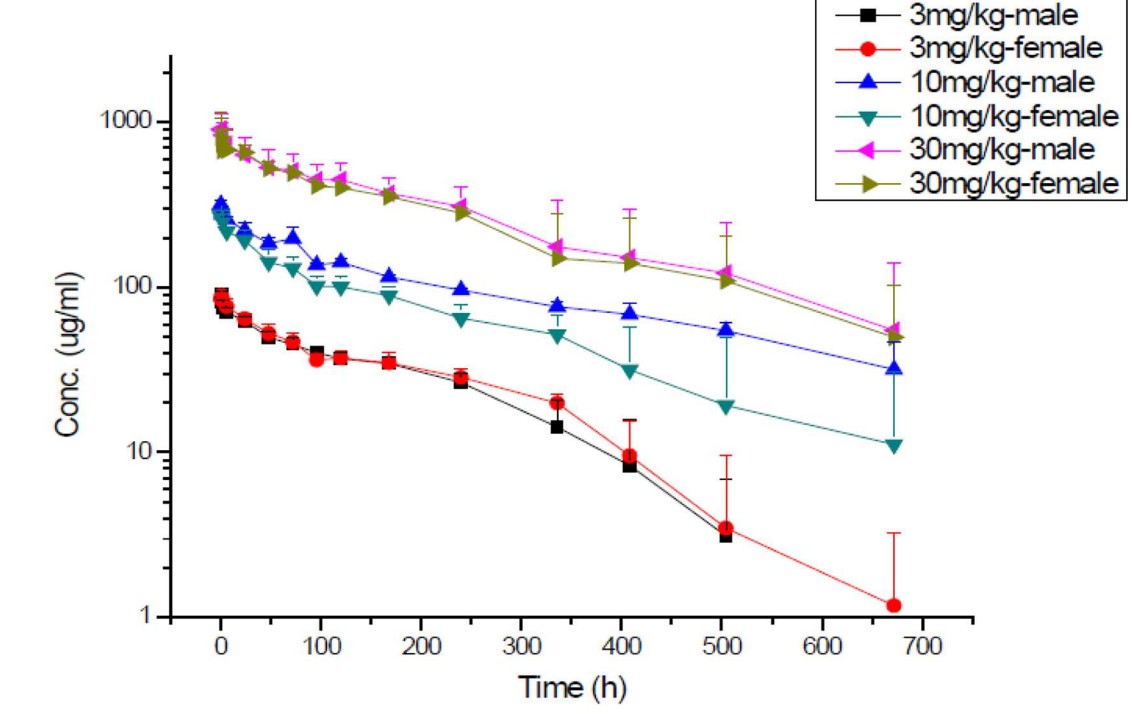

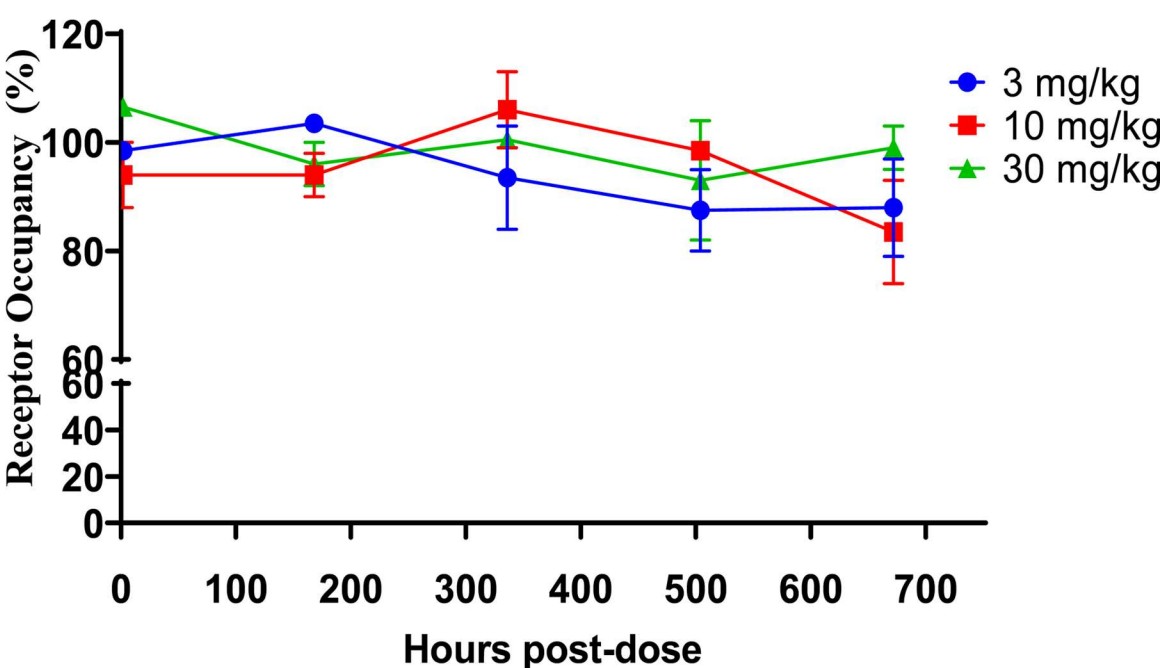

**Fig 12. PD-1 receptor occupancy and PK profiles.** (a) Mean Serum drug concentration-time curves of HLX10 following a single IV-infusion at 3, 10 and 30 mg/kg (N = 3). (b) Receptor occupancy rate of HLX10 on PD-1 on the surface of T cells following a single IV-infusion at 3, 10 and 30 mg/kg, which was assessed by flow cytometry.

therapeutic mAbs targeting PD-1 might still be warranted for multiple tumors treatments. In this article, we compare the early preclinical and the molecular characteristics of HLX10 to Nivolumab and Pembrolizumab. HLX10 is a fully humanized anti-PD-1 monoclonal antibody that binds to human PD-1 expressed on CHO-S cells and T-cells. Moreover, HLX10 shows potent PD-L1 and PD-L2 blocking activity in recombinant protein blocking and reporter cell assays and was at least as efficient as Nivolumab reference antibody in enhancing T-cell responses and cytokine production *in vitro*. No ADCC or CDC activity was observed with HLX10 when using syngeneic PD-1-expressing activated T-cells as target T-cells. These results suggest that HLX10 is unlikely to deplete PD-1-positive cells in patients. Lack of HLX10-mediated ADCC or CDC activity is consistent with the expected lack of effector function of the IgG4 Fc region, as observed in Nivolumab. The anti-tumor activity of HLX10 was demonstrated in several models, including, syngeneic EMT-6 and MC38 models, and xenograft hPBMC co-mixture models, exhibiting up 51% TGI in EMT-6, >80% TGI in MC38, up to 53% TGI in HT-29, and 60% TGI in NCI-292 models. Additionally, HLX10's activity was more pronounced than Nivolumab in a dose response study in NCI-292 model. In hu-CD34$^+$ NSG$^{TM}$ mice implanted orthotopically with the TNBC MDA-MD-231 cell line, HLX10 showed equivalent anti-tumor activity as Pembrolizumab. Additionally, HLX10 enhanced tumor shrinkage when combined with anti-angiogenic antibody HLX04. Many studies have shown that blocking angiogenesis can increase the efficacy of immune checkpoint PD-1/PD-L1 inhibitors [24]. The possible mechanisms of action include: (1) VEGF-A secreted by tumor cells in the tumor microenvironment induces the expression of PD-1 and other immune check-point inhibitory molecules on CD8$^+$ T-cells, which is related to the exhaustion of T-cells within tumors [26]; (2) Anti-angiogenic agents can normalize the blood vessel and increased tumor oxygenation through reducing tumor blood vessel distribution, compactness, tight junctions, or pressures and result in normalized blood vessels as well as increased tumor oxygenation, which in turn increases the infiltration and stimulation of CD4$^+$ and CD8$^+$ T-cells within tumors.

Mechanistically, structural analysis of PD-1-HLX10-Fab complex suggests that HLX10 competes with PD-L1 binding in similar fashion to the approved Nivolumab and

**Table 4. Pharmacokinetic parameters for HLX10 in cynomolgus monkeys.**

| Dose mg/kg | Sex | | $t_{1/2}$ | $C_{max}$ | $AUC_{last}$ | $AUC_{inf}$ | Vz | CL | MRT | $C_{max}$ | $AUC_{last}$ |
|---|---|---|---|---|---|---|---|---|---|---|---|
| | | | h | μg/mL | h*mg/mL | h*mg/mL | mL/kg | mL/h/kg | h | Ratio | Ratio |
| 3 | M | Mean | 137.97 | 91.33 | 13.25 | 14.38 | 41.18 | 0.21 | 153.02 | 1.00 | 1.00 |
| | | SD | 41.80 | 3.10 | 1.01 | 0.99 | 10.02 | 0.01 | 27.65 | | |
| | F | Mean | 159.21 | 89.07 | 14.45 | 15.75 | 43.56 | 0.19 | 169.49 | 1.00 | 1.00 |
| | | SD | 40.49 | 3.26 | 0.43 | 1.31 | 9.84 | 0.02 | 36.78 | | |
| 10 | M | Mean | 256.99 | 318.64 | 63.33 | 75.92 | 48.47 | 0.13 | 231.28 | 3.49 | 4.78 |
| | | SD | 56.13 | 20.86 | 2.79 | 9.70 | 4.43 | 0.02 | 15.66 | | |
| | F | Mean | 173.29 | 275.05 | 41.06 | 46.43 | 53.52 | 0.23 | 181.31 | 3.09 | 2.84 |
| | | SD | 77.62 | 18.14 | 9.35 | 13.42 | 14.00 | 0.07 | 47.54 | | |
| 30 | M | Mean | 191.69 | 922.95 | 173.02 | 201.61 | 38.05 | 0.17 | 199.68 | 10.11 | 13.05 |
| | | SD | 159.95 | 232.81 | 52.80 | 99.25 | 18.89 | 0.07 | 74.55 | | |
| | F | Mean | 178.78 | 862.52 | 160.81 | 179.56 | 39.21 | 0.18 | 194.16 | 9.68 | 11.13 |
| | | SD | 120.96 | 246.71 | 34.06 | 53.69 | 20.32 | 0.06 | 73.85 | | |

Note: $C_{max}$ ratio and $AUC_{last}$ ratio are relative to the 3 mg/kg dose group data.

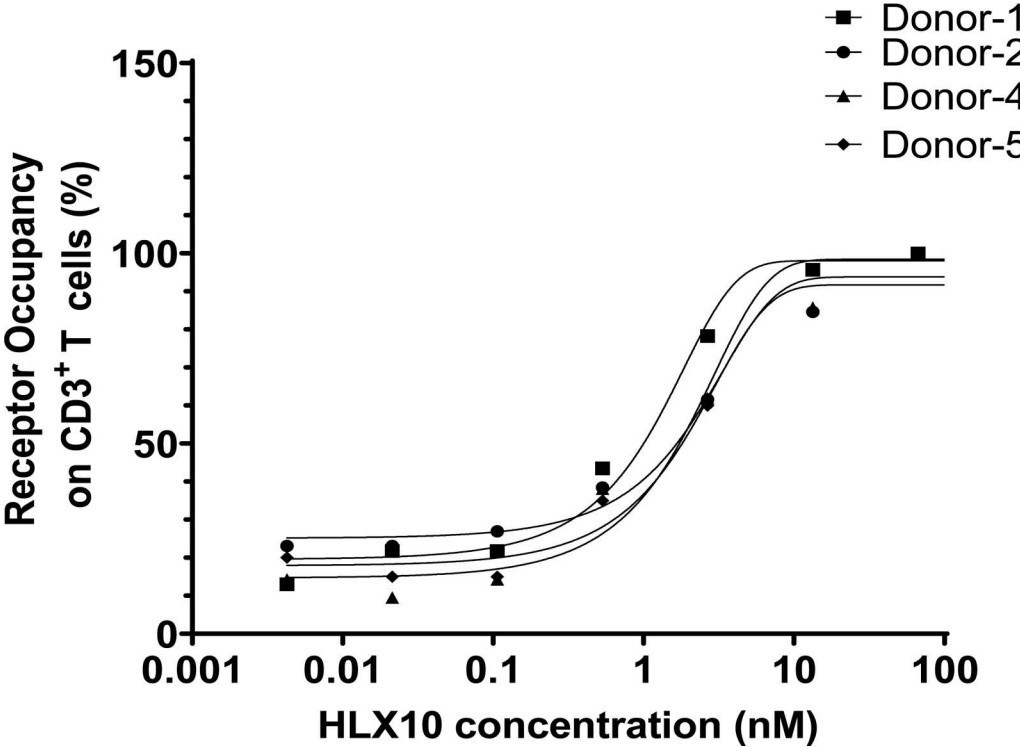

**Fig 13. Human PD-1 receptor occupancy.** *In vitro* receptor occupancy rate of HLX10 on PD-1 on the surface of human activated T cells, which was assessed by flow cytometry.

Pembrolizumab mAbs. HLX10's epitope overlaps with Pembrolizumab more than Nivolumab. Interestingly, both HLX10 and Pembrolizumab share binding to C'D loop in hPD-1. Furthermore, Arg86 was identified as a key residue for HLX10 binding forming several contacts with resides in HCDR3 and HCDR1.

HLX10 binds to cynomolgus monkey PD-1 ECD but not to the ECD of mouse or rat. The species cross-reactivity data supports the selection of cynomolgus monkey as the most relevant species for safety evaluation of HLX10. PK analysis in cynomolgus monkey showed that HLX10 followed a pharmacokinetics typical of a recombinant monoclonal antibody, with small volume of distribution and long elimination half-life. Moreover, RO studies suggested that the majority of PD-1 can be saturated at 2 μg/mL loading dose, providing an estimate for phase I dose selection. Moreover, in subsequent toxicity studies, HLX10 was well tolerated when administered at up to 100 mg/kg for 4 weeks, which was determined as a No Observed Adverse Effect Level (NOAEL). Therefore, HLX10 is well positioned to provide alternative treatment options for cancer patients. HLX10 is currently studied in phase III trials (NCT04063163) in combination with chemotherapy as first-line treatment of squamous non-small cell lung cancer (sqNSCLC) and HLX04 (a biosimilar of Avastin) as first-line treatment of non-squamous NSCLC in phase I trial (NCT03757936).

## Supporting information

**S1 Fig.** a. BLI binding kinetics. b. SPR binding kinetics of HLX10. c. SPR binding kinetics of Nivolumab.
(PPTX)

**S2 Fig. MC-38 in vivo model.**
(PPTX)

**S3 Fig. NCI-H292/hPBMC co-mixture model.**
(PPTX)

**S4 Fig. HLX10 and PD-1 contact.**
(PPTX)

**S1 File. Supplemental material and methods.**
(DOCX)

**S2 File. Response to non-human primate question.**
(DOCX)

## Author Contributions

**Conceptualization:** Hassan Issafras, Chi-Ling Tseng, Lisa Xiao, Ya-Chin Hsiao, Eugene Liu, Weidong Jiang.

**Data curation:** Hassan Issafras, Shilong Fan, Chi-Ling Tseng, Yunchih Cheng, Peihua Lin, Lisa Xiao, Yun-Ju Huang, Chih-Hsiang Tu.

**Formal analysis:** Shilong Fan, Chi-Ling Tseng, Peihua Lin, Chih-Hsiang Tu, Ya-Chin Hsiao, Yen-Hsiao Chen, Yi-Ting Mao.

**Investigation:** Chien-Hsin Ho, Ou Li, Eric Zhang, Yi-Ting Mao.

**Methodology:** Chi-Ling Tseng, Yunchih Cheng, Peihua Lin, Lisa Xiao, Yun-Ju Huang, Chih-Hsiang Tu, Ya-Chin Hsiao, Min Li, Yen-Hsiao Chen, Chien-Hsin Ho, Yanling Wang, Sandra Chen, Zhenyu Ji.

**Supervision:** Weidong Jiang.

**Visualization:** Chih-Hsiang Tu.

**Writing – original draft:** Hassan Issafras.

**Writing – review & editing:** Hassan Issafras, Shilong Fan, Chi-Ling Tseng, Shumin Yang, Weidong Jiang.

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
