## [Decision Letter · Decision Letter 0]

14 May 2021

PONE-D-21-10696

Structural basis of HLX10 PD-1 receptor recognition, a promising anti-PD-1 antibody clinical candidate for cancer immunotherapy

PLOS ONE

Dear Dr. Issafras,

Thank you for submitting your manuscript to PLOS ONE. After careful consideration, we feel that it has merit but does not fully meet PLOS ONE’s publication criteria as it currently stands. Therefore, we invite you to submit a revised version of the manuscript that addresses the points raised during the review process.

We look forward to receiving your revised manuscript.

Kind regards,

Irina V. Balalaeva, PhD

Academic Editor

PLOS ONE

Journal Requirements:

2.  At this time, we request that you  please report additional details in your Methods section regarding animal care, as per our editorial guidelines: 1) Please provide details of animal welfare (e.g., shelter, food, water, environmental enrichment) 2) Please describe any steps taken to minimize animal suffering and distress, such as by administering anesthetics or analgesics, 3) Please include the method of sacrifice, 4) Please include the maximum sizes tumors grew to in your experiments and 5) Please describe the post-operative and post tumor implantation care received by the animals, including the frequency of monitoring and the criteria used to assess animal health and well-being. Thank you for your attention to these requests.

"The author(s) received no specific funding for this work.

Hassan Issafras, Chi-Ling Tseng, Yunchih Cheng, Peihua Lin, Lisa Xiao, Yun-Ju Huang, Chih-Hsiang Tu, Ya-Chin Hsiao,  Yen-Hsiao Chen, Chien-Hsin Ho, Ou Li, Yanling Wang, Sandra Chen, Zhenyu Ji, Eric Zhang, Yi-Ting Mao, Eugen Liu, Shumin Yang and Weidong Jiang were employees of either Hengenix Inc or Shanghai Henlius Biotech, Inc., P. R. China.

Shilong Fan and Min Li received funding from Shanghai Henlius Biotech, Inc."

We note that one or more of the authors have an affiliation to the commercial funders of this research study : Hengenix Inc., Shanghai Henlius Biotech, Inc., HanchorBio Inc., Ltd, Anwita Biosciences.

3.1. Please provide an amended Funding Statement declaring this commercial affiliation, as well as a statement regarding the Role of Funders in your study. If the funding organization did not play a role in the study design, data collection and analysis, decision to publish, or preparation of the manuscript and only provided financial support in the form of authors' salaries and/or research materials, please review your statements relating to the author contributions, and ensure you have specifically and accurately indicated the role(s) that these authors had in your study. You can update author roles in the Author Contributions section of the online submission form.

3.2. Please also provide an updated Competing Interests Statement declaring this commercial affiliation along with any other relevant declarations relating to employment, consultancy, patents, products in development, or marketed products, etc.  

7. Please amend the manuscript submission data (via Edit Submission) to include author Chi-Ling Tseng, Yunchih Cheng, Peihua Lin, Lisa Xiao, Yun-Ju Huang, Chih-Hsiang Tu, Ya-Chin Hsiao, Min Li, Yen-Hsiao Chen3, Chien-Hsin Ho, Ou Li, Yanling Wang, Sandra Chen, Zhenyu Ji, Eric Zhang, Yi-Ting Mao, Eugen Liu, Shumin Yang.

8. We note that you have included the phrase “data not shown” in your manuscript. Unfortunately, this does not meet our data sharing requirements. PLOS does not permit references to inaccessible data. We require that authors provide all relevant data within the paper, Supporting Information files, or in an acceptable, public repository. Please add a citation to support this phrase or upload the data that corresponds with these findings to a stable repository (such as Figshare or Dryad) and provide and URLs, DOIs, or accession numbers that may be used to access these data. Or, if the data are not a core part of the research being presented in your study, we ask that you remove the phrase that refers to these data.

Reviewers' comments:

Reviewer's Responses to Questions

**Comments to the Author**

1. Is the manuscript technically sound, and do the data support the conclusions?

Reviewer #1: Yes

Reviewer #2: Yes

Reviewer #3: Yes

2. Has the statistical analysis been performed appropriately and rigorously? 

Reviewer #1: Yes

Reviewer #2: Yes

Reviewer #3: Yes

3. Have the authors made all data underlying the findings in their manuscript fully available?

Reviewer #1: Yes

Reviewer #2: Yes

Reviewer #3: Yes

4. Is the manuscript presented in an intelligible fashion and written in standard English?

Reviewer #1: Yes

Reviewer #2: Yes

Reviewer #3: Yes

5. Review Comments to the Author

Reviewer #1: The authors present an extensive study describing the generation of an anti-PD-1 IgG4 monoclonal antibody with a remarkable activity in vitro (binding to PD-1, to PD-1 expressing cells, inhibition of PD-1/PD-L1 binding, activity on humanT cells) and in vivo in several tumor models. Also a pharmacokinetic study in cynomolgus monkeys has been performed is here reported. The antibody named HLX10 appears as a promising new candidate for immunotherapy.

The work is well designed, well executed and well presented and can be accepted with minor revisions.

A more detailed description of the phage display library constructiuon and screening should be provided in terms of complexity, panning rounds, number of new sequences etc.

Reviewer #2: In this paper, author reported that HLX10, a novel fully humanized anti PD-1 IgG4 mAb, blocked PD1/PD-L1 interaction and activated T-cell proliferation and cytokine secretion in vitro. HLX10 demonstrated significant antitumor efficacy in several in 5 mouse models and synergized with Avastin biosimilar to promote robust tumor activity. Finally, author showed a co-crystal structure of the antigen-binding fragment (Fab) of HLX10 in complex with PD-1 at a 1.78-Å resolution. Besides, HLX10 was well tolerated by Monkey single dose pharmacokinetics assay. Altogether, comprehensive study proved that HLX10 had better bioactivity to FDA approved nivolumab and pembrolizumab.

Minor questions:

1. As author described “The EMT-6 is considered an anti-PD-1 resistant tumor, due to an immune excluded phenotype, macrophage-derived suppressive infiltration, and epithelial-to-mesenchymal transition (EMT) [22, 23]. Upon treatment of established EMT-6/ tumors in BALB/c mice, HLX10 was able to significantly inhibit tumor growth with some responding and nonresponding mice. Please add a short discussion. You may refer recent study on anti-PD-L1/TGF-b bispecific antibody (Yi M, et al. The construction, expression, and enhanced anti-tumor activity of YM101: a bispecific antibody simultaneously targeting TGF-β and PD-L1. J Hematol Oncol, 2021;14:27).

2. HLX10 was shown to synergize with Avastin biosimilar to promote robust tumor activity. For background information as to the combination of ICI with anti-angiogenesis, please refer recent review and make short discussion (Yi M, et al. Synergistic effect of immune checkpoint blockade and anti-angiogenesis in cancer treatment. Mol Cancer, 2019; 19:60).

3. Could you please show CD8+ T-cells and blood vessel for tumor tissues from in vivo study of Fig 8A, B?

Reviewer #3: The research article by Hassan Issafras, Shilong Fan et al. deals with the structural, biological and pharmacological characterizations of HLX10, a humanized IgG4 monoclonal antibody against PD-1 receptor.

A comparative analysis of binding activity of HLX10 and several approved mAb to PD-1 was carried out, showing a similar efficiency of HLX10 to the reference antibody Nivolumab in enhancing T-cell responses and cytokine production in vitro. The anticancer activity of HLX10 was then demonstrated using several syngeneic and xenograft models. Moreover, an increase of antitumor efficacy of HLX10 was observed when used in combination with HLX10 enhanced an anti-angiogenic antibody (HLX04).

Crystallographic studies of HLX10/PD-1 complex disclose a similar binding mode to the approved Pembrolizumab mAb and identify Arg86 of PD-1 as key residue for HLX10 binding. PK studies in cynomolgus monkey showed that HLX10 is endowed with a small volume of distribution and long elimination half-life, which is common to recombinant monoclonal antibodies. Toxicity studies show also that HLX10 is well tolerated up to 100 mg/kg for 4 weeks. Overall, the work is well conceived, the English language is very good, and results are of interest to researchers working in the field of cancer immunotherapies. The methodological part and experiments are well executed and compliant to a high technical standard. Conclusions are appropriately written and supported by the results.

However, before accepting the paper for publication in PLOS One, authors should address the following minor issues.

- Line 303-305: Authors should mention the technique used to evaluate HLX10 binding to PD-1 in CHO transfected cells (in the same way as it is reported for the BLI assay to check the monomeric binding affinity).

- Line 412: The text ‘Error! Reference source not found’ appears, the author should insert the citation properly.

- Line 428: Authors should mention that crystallographic methods are reported in supplementary Material

- The legend and clarity of figure 6 should be improved. It is indeed unclear the meaning of the nine

growth curves (solid black lines) representing HLX10 15 mg/kg. Do they show the growth of EMT breast cancer syngeneic tumors following the injections (6 or 9 injections) ?

- In the legend of figure 9 it is stated that “Arg86 of hPD-1 is involved in several hydrogen bonds, salt bridge and hydrophobic interaction with HLX10.” Being a polar positively charged residue, Arg86 cannot form hydrophobic interactions with HLX10.

- In the conclusion part, authors should specify that the combination therapy HLX10/HLX04 (NCT03757936) is a phase I clinical study.

6. PLOS authors have the option to publish the peer review history of their article (what does this mean?). If published, this will include your full peer review and any attached files.

Reviewer #1: **Yes: **Menotti Ruvo

Reviewer #2: No

Reviewer #3: **Yes: **Antonio Macchiarulo

---

## [Author Response · Author response to Decision Letter 0]

22 Aug 2021

Dear Editor,

We thank the reviewers for their insightful comments on the manuscript and have edited the manuscript to address their concerns. We submitted 2 versions: final version and one with track changes.

We believe that the manuscript is now suitable for publication in PLOS.

We also modified some items in the manuscript to satisfy the editorial guidelines. The followings are the answers to the reviewer’s comments/questions and the list of edits made.

Reviewer #1: The authors present an extensive study describing the generation of an anti-PD-1 IgG4 monoclonal antibody with a remarkable activity in vitro (binding to PD-1, to PD-1 expressing cells, inhibition of PD-1/PD-L1 binding, activity on humanT cells) and in vivo in several tumor models. Also a pharmacokinetic study in cynomolgus monkeys has been performed is here reported. The antibody named HLX10 appears as a promising new candidate for immunotherapy.

The work is well designed, well executed and well presented and can be accepted with minor revisions.

A more detailed description of the phage display library construction and screening should be provided in terms of complexity, panning rounds, number of new sequences etc.

Thank you for the comments. We have added more details about the phage library construction, and screening etc. to the material and method section.

Reviewer #2: In this paper, author reported that HLX10, a novel fully humanized anti PD-1 IgG4 mAb, blocked PD1/PD-L1 interaction and activated T-cell proliferation and cytokine secretion in vitro. HLX10 demonstrated significant antitumor efficacy in several in 5 mouse models and synergized with Avastin biosimilar to promote robust tumor activity. Finally, author showed a co-crystal structure of the antigen-binding fragment (Fab) of HLX10 in complex with PD-1 at a 1.78-Å resolution. Besides, HLX10 was well tolerated by Monkey single dose pharmacokinetics assay. Altogether, comprehensive study proved that HLX10 had better bioactivity to FDA approved nivolumab and pembrolizumab.

Minor questions:

1. As author described “The EMT-6 is considered an anti-PD-1 resistant tumor, due to an immune excluded phenotype, macrophage-derived suppressive infiltration, and epithelial-to-mesenchymal transition (EMT) [22, 23]. Upon treatment of established EMT-6/ tumors in BALB/c mice, HLX10 was able to significantly inhibit tumor growth with some responding and nonresponding mice. Please add a short discussion. You may refer recent study on anti-PD-L1/TGF-b bispecific antibody (Yi M, et al. The construction, expression, and enhanced anti-tumor activity of YM101: a bispecific antibody simultaneously targeting TGF-β and PD-L1. J Hematol Oncol, 2021;14:27).

2. HLX10 was shown to synergize with Avastin biosimilar to promote robust tumor activity. For background information as to the combination of ICI with anti-angiogenesis, please refer recent review and make short discussion (Yi M, et al. Synergistic effect of immune checkpoint blockade and anti-angiogenesis in cancer treatment. Mol Cancer, 2019; 19:60).

3. Could you please show CD8+ T-cells and blood vessel for tumor tissues from in vivo study of Fig 8A, B?

Thank you for the comments and questions. We have added shorth discussions to the results section, and we have added the references suggested. 

We agree elucidating the synergy between ICI and anti-angiogenesis inhibitors is an important question and would have been nice to show during these studies. We referenced (Yi M, et al. Synergistic effect of immune checkpoint blockade and anti-angiogenesis in cancer treatment. Mol Cancer, 2019; 19:60) review, which will add some background and general context. 

Unfortunately, CD8+ and blood vessel were not examined in these studies as they were not the primary goals of the experiment. 

Reviewer #3: The research article by Hassan Issafras, Shilong Fan et al. deals with the structural, biological and pharmacological characterizations of HLX10, a humanized IgG4 monoclonal antibody against PD-1 receptor.

A comparative analysis of binding activity of HLX10 and several approved mAb to PD-1 was carried out, showing a similar efficiency of HLX10 to the reference antibody Nivolumab in enhancing T-cell responses and cytokine production in vitro. The anticancer activity of HLX10 was then demonstrated using several syngeneic and xenograft models. Moreover, an increase of antitumor efficacy of HLX10 was observed when used in combination with HLX10 enhanced an anti-angiogenic antibody (HLX04).

Crystallographic studies of HLX10/PD-1 complex disclose a similar binding mode to the approved Pembrolizumab mAb and identify Arg86 of PD-1 as key residue for HLX10 binding. PK studies in cynomolgus monkey showed that HLX10 is endowed with a small volume of distribution and long elimination half-life, which is common to recombinant monoclonal antibodies. Toxicity studies show also that HLX10 is well tolerated up to 100 mg/kg for 4 weeks. Overall, the work is well conceived, the English language is very good, and results are of interest to researchers working in the field of cancer immunotherapies. The methodological part and experiments are well executed and compliant to a high technical standard. Conclusions are appropriately written and supported by the results.

However, before accepting the paper for publication in PLOS One, authors should address the following minor issues.

- Line 303-305: Authors should mention the technique used to evaluate HLX10 binding to PD-1 in CHO transfected cells (in the same way as it is reported for the BLI assay to check the monomeric binding affinity).

The flow cytometry method used to evaluate HLX10 binding to PD-1 in CHO transfected cells is described in supplemental MM

- Line 412: The text ‘Error! Reference source not found’ appears, the author should insert the citation properly.

Error corrected 

- Line 428: Authors should mention that crystallographic methods are reported in supplementary Material

Added

- The legend and clarity of figure 6 should be improved. It is indeed unclear the meaning of the nine

growth curves (solid black lines) representing HLX10 15 mg/kg. Do they show the growth of EMT breast cancer syngeneic tumors following the injections (6 or 9 injections) ?

Each black curve represents an individual animal (N=10). The dotted line represents the average of 10 animals from the vehicle group. We have added additional description to the legend to make the figures clearer. The shaded area is the treatment window (6 injections). We hope this will be satisfactory.

- In the legend of figure 9 it is stated that “Arg86 of hPD-1 is involved in several hydrogen bonds, salt bridge and hydrophobic interaction with HLX10.” Being a polar positively charged residue, Arg86 cannot form hydrophobic interactions with HLX10.

We corrected this error.

- In the conclusion part, authors should specify that the combination therapy HLX10/HLX04 (NCT03757936) is a phase I clinical study.

We have added this info

---

## [Editor Report · Decision Letter 1]

15 Sep 2021

Structural basis of HLX10 PD-1 receptor recognition, a promising anti-PD-1 antibody clinical candidate for cancer immunotherapy

PONE-D-21-10696R1

Dear Dr. Issafras,

We’re pleased to inform you that your manuscript has been judged scientifically suitable for publication and will be formally accepted for publication once it meets all outstanding technical requirements.

Kind regards,

Irina V. Balalaeva, PhD

Academic Editor

PLOS ONE
---

## [Editor Report · Acceptance letter]

18 Nov 2021

PONE-D-21-10696R1 

Structural basis of HLX10 PD-1 receptor recognition, a promising anti-PD-1 antibody clinical candidate for cancer immunotherapy 

Dear Dr. Issafras:

I'm pleased to inform you that your manuscript has been deemed suitable for publication in PLOS ONE. Congratulations! Your manuscript is now with our production department. 

Kind regards, 

on behalf of

Dr. Irina V. Balalaeva 

Academic Editor

PLOS ONE